# Quantifying the seasonal variations and regional transport of PM$_{2.5}$ in the Yangtze River Delta region, China: Characteristics, sources, and health risks

Yangzhihao Zhan[a], Min Xie[a,b], Wei Zhao[c], Tijian Wang[a], Da Gao[d], Pulong Chen[e], Jun Tian[f], Kuanguang Zhu[g], Shu Li[a], Bingliang Zhuang[a], Mengmeng Li[a], Yi Luo[a], Runqi Zhao[a]

[a] School of Atmospheric Sciences, Nanjing University, Nanjing 210023, China
[b] School of Environment, Nanjing Normal University, Nanjing 210023, China
[c] Nanjing Institute of Environmental Sciences, Ministry of Ecology and Environment of the People's Republic of China,Nanjing 210023, China
[d] State Key Joint Laboratory of Environment Simulation and Pollution Control, School of Environment, Tsinghua University, Beijing 100084, China
[e] Net Zero Era (Jiangsu) Environmental Technology Co., Nanjing 210023, China
[f] Academy of Environmental Planning and Design. Co.,Ltd., Nanjing University, Nanjing 210023, China
[g] Hubei Provincial Academy of Eco-Environmental Sciences, Wuhan 430073, China

*Correspondence to*: Min Xie (minxie@nju.edu.cn), Wei Zhao (zhaowei@nies.org)

**Abstract.** Given the increasing complexity of the chemical composition of PM$_{2.5}$, identifying and quantitatively assessing the contributions of pollution sources has played an important role in formulating policies to control particle pollution. This study provides a comprehensive assessment between PM$_{2.5}$ chemical characteristics, sources, and health risks based on sampling data conducted over one year (March 2018 to February 2019) in Nanjing. Results show that PM$_{2.5}$ exhibits a distinct variation across different seasons, which is primarily driven by emissions, meteorological conditions, and chemical conversion of gaseous pollutants. First, the chemical mass reconstruction shows that secondary inorganic aerosols (SIA, 62.5 %) and carbonaceous aerosols (21.3%) contributed most to the PM$_{2.5}$ mass. The increasing oxidation rates of SO$_2$ and NO$_2$ from summer to winter indicate that the secondary transformation of gaseous pollutants is strongly positively correlated with relative humidity. Second, the positive matrix factorization (PMF) method shows that identified PM$_{2.5}$ sources include secondary inorganic aerosol sources (SIS, 42.5%), coal combustion (CC, 22.4%), industry source (IS, 17.3%), vehicle emission (VE, 10.7%), fugitive dust (FD, 5.8%) and other sources (1.3%). The Hybrid Single Particle Lagrangian Integrated Trajectory (HYSPLIT) model and the concentration-weighted trajectory (CWT) analysis are used to further explore different spatial distributions and regional transport of sources. The concentrations (10-11 μg·m$^{-3}$) of SIS and CC distribute in Nanjing and central China in winter. The concentrations (8-10 μg·m$^{-3}$) of IS and VE are potentially located north of Jiangsu, Anhui, and Jiangxi. Finally, the health risk assessment indicates that the carcinogenic and non-carcinogenic risks of toxic elements (Cr, As, Ni, Mn, V, and Pb) mainly come from IS, VE, and CC, which are within the tolerance or acceptable level. Although the main source of pollution in Nanjing is SIS at present, we should pay more attention to the health burden of vehicle emissions, coal combustion, and industrial processes.

34

## 1. Introduction

PM$_{2.5}$ is particulate matter with an aerodynamic equivalent diameter less than or equal to 2.5 μm, and one of the most important air pollutants, which can affect air quality (Sharma et al., 2020), atmospheric visibility (Tseng et al., 2019) and ecosystems (Li et al., 2021). PM$_{2.5}$ can directly enter the human body through the respiratory system and lead to increased health risks (Kumar et al., 2019; Sulaymon et al., 2021). PM$_{2.5}$ concentrations in the United States and Europe have begun to decrease since the 1980s, and those in Japan gradually decreased after 2012 (Zhang et al., 2020). In China, the annual average concentration of PM$_{2.5}$ has decreased by 50% with the implementation of the Air Pollution Prevention and Control Action Plan (APPCAP) in 2013. However, annual PM$_{2.5}$ concentrations in most cities are greater than 10 μg·m$^{-3}$, the air quality guideline of the World Health Organization (Song et al., 2017; Zeng et al., 2019; Cheng et al., 2021), and the number of deaths caused by PM$_{2.5}$ exceeds one million per year (Zhu et al., 2020). It indicates that a comprehensive assessment between PM$_{2.5}$ chemical characteristics, sources, and health risks is significant for pollution control measures in the key regions of China.

Understanding the chemical composition of PM$_{2.5}$ is important for formulating control strategies. Sulfate, nitrate, and ammonium (SNA) are the major secondary inorganic aerosols, whose chemical conversion occurs in homogeneous and heterogeneous reactions (Fan et al., 2020; Chow et al., 2022). Variations in the form of SO$_4^{2-}$ and NH$_3$ lead to variations in the acid-base balance of aerosols (Roper et al., 2019). Organic carbon (OC) comprises thousands of organic compounds. Elemental carbon (EC) is stable and mainly derived from primary sources of combustion products (Wu et al., 2020; Zhang et al., 2022). Both NO$_3^-$/SO$_4^{2-}$ and OC/EC ratios can be reasonably used to evaluate the contribution of mobile and stationary sources to PM$_{2.5}$ in the atmosphere (Zhan et al., 2021). To identify the sources of PM$_{2.5}$, receptor modelings have been developed, which include positive matrix factorization (PMF), chemical mass balance (CMB), and principal component analysis (PCA) (Zong et al., 2016; Lv et al., 2020). Recently, the combination of the PMF model and trajectory modeling has proven to be powerful to identify source regions and quantify chemical compositions for a receptor site (Zheng et al., 2019). Air exposure models have been widely used to compare the health outcomes of people exposed to different levels of air pollution (Thurston et al., 2016; Conibear et al., 2018). Long-term exposure to PM$_{2.5}$ is particularly significant for cardiovascular disease mortality (Hayes et al., 2020). Trace metals (Cr, Ni, Mn, V, and Pb) are a minor component of PM$_{2.5}$ in qualitative terms, but the health risk of toxic elements through inhalation of PM$_{2.5}$ exceeds acceptable levels (Jiang et al., 2018; Jeong et al., 2019; Xie et al., 2020). Health risk assessments have been widely used to assess further the non-carcinogenic and carcinogenic health risks of toxic elements in PM$_{2.5}$ (Behrooz et al., 2021; Fang et al., 2021; Li et al., 2022).

Chemical characteristics of PM$_{2.5}$ have been widely investigated in the Beijing-Tianjin-Hebei (BTH), the Yangtze River Delta (YRD), and the Pearl River Delta (PRD) during the last decade (Huang et al., 2017; Liu et al., 2017; Li et al., 2020). In the megacity of China, the occurrence of haze may be exacerbated by interactions between aerosols and meteorological conditions and regional transport (Zeng et al., 2019; Fan et al., 2020; Wang et al., 2023). The YRD region is China's scientific research base and comprehensive transportation hub. The annual PM$_{2.5}$ concentration in the YRD has been reduced

by 45.6% from 2016 to 2018. However, as a mega-city in the YRD, the PM$_{2.5}$ in Nanjing still exceeds the National Ambient
Air Quality Standard (35 μg·m$^{-3}$ as an annual average) by more than 38 % (Nie et al., 2018). Source apportionment studies
mainly focus on the relative importance of local emission and regional transportation on PM$_{2.5}$ at a specific site using the
PMF model and the backward trajectory analysis (Zheng et al., 2019; Yan et al., 2021; Lv et al., 2022). Some studies
involved the health risks of toxic elements in PM$_{2.5}$ (Zhang et al., 2019; Fang et al., 2021), and only a few studies discussed
the classification of toxic elements according to PMF results (Wang et al., 2019; Wang et al., 2020). However, there were
two shortcomings in previous studies: (1) Given the uneven geographical distribution of observation sites and difficulties in
data collection, most studies were based on short-term data comparisons and lacked systematic comparisons of the
distinctive seasonality, regional transport, and meteorological effects of various elements and sources. (2) A comprehensive
assessment of the health risks of toxic elements in each source of PM$_{2.5}$ was still scarce, which limited the implementation of
long-term pollution control measures in megacities.
In this work, we provide high-quality composition data for PM$_{2.5}$ in the typical YRD city, including their chemical
characteristics and diurnal variations. Besides, the measured PM$_{2.5}$ in its entirety is successfully apportioned to various
contributing sources by PMF and CWT methods. Finally, potential risks associated with exposure to airborne toxic elements
are identified based on the health risk assessment. The results can systematically assess the relationship between chemical
characteristics, sources, and health risks of PM$_{2.5}$, and serve to guide PM$_{2.5}$ control measures for other megacities.

**2. Data and Methodology**
**2.1 Chemical component sampling, air quality and meteorological data**
Hourly concentrations of particulate matter (PM) components from December 2018 to February 2019 in Nanjing were
used in this study. PM$_{2.5}$ samples were collected on the rooftop of the School of Atmospheric Sciences, Xianlin Campus,
Nanjing University (32.12 °N, 118.96 °E). The elemental carbon (EC), organic carbon (OC), 30 trace elements, and 8
soluble components in aerosols were quantified in each PM$_{2.5}$ sample.
EC and OC samples were analyzed by the online carbon fraction Monitor (EA-32, Everisetech Co., Beijing). Taking
advantage of the fact that EC was more difficult to be volatilized than OC, the instrument separated OC and EC by step
heating, catalyzed sequentially, and then determined by the non-dispersive infrared method.
30 trace elements included Si, Al, As, Ca, K, Co, Mo, Ag, Sc, Tl, Pd, Br, Te, Ga, Cs, Pb, Se, Hg, Cr, Cd, Zn, Cu, Ni, Fe,
Mn, Ti, Sb, Sn, and V. The components of trace elements were collected by the atmospheric heavy metal Monitor (AMS-100,
Fpigroup Co., Hangzhou). We used a particle cutting head to collect particles with an aerodynamic equivalent diameter of
less than 100/10/2.5 μm in the ambient air, used organic microporous filter membranes to enrich the collected particles, used
the principle of β-ray absorption to detect the concentration of particles enriched on the filter membranes and used the
principle of X-ray Fluorescence to detect the concentration of more than 30 types of trace elements in the particles (Wang et
al., 2020).

101 8 soluble components included $Na^+$, $K^+$, $Mg^{2+}$, $Ca^{2+}$, $Cl^-$, $NO_3^-$, $SO_4^{2-}$, and $NH_4^+$. The soluble components sampling

102 instrument was the In-situ Gas and Aerosol Compositions Monitor (IGAC, Fortelice International Co., Taiwan). It consisted

103 of the wet concentric circular tube, the gas gel processor, and the ion chromatograph. The sampling inlet was about 20 m

104 above the ground and the flow rate was 16.67 L·min$^{-1}$. The collected liquid samples were filtered by defoaming and then

105 injected into the ion chromatography analyzers to analyze the ion components from the gases and the aerosols. The detection

106 limits were below 0.12 μg·m$^{-3}$ and the collection efficiency was higher than 90% (Zhan et al., 2021).

107 Air pollutants, including $PM_{2.5}$, $PM_{10}$, $O_3$, $NO_2$, $SO_2$, and CO, were monitored by the National Environmental

108 Monitoring Center (NEMC) of China. The nationwide observation network began operating in 74 major cities in 2013, and it

109 included 1597 nonrural sites covering 454 cities by 2017 (Gao et al., 2021). The monitoring Xianlin Station (32.10 °N,

110 118.93 °E) collected air pollutant data and automatically measured hourly air pollutants. These data were issued hourly on

111 the national urban air quality real-time publishing platform (https://air.cnemc.cn:18007/, last access: 7 April 2023).

112 Meteorological parameters included air pressure, air temperature, relative humidity, wind speed, and boundary layer height.

113 We collected hourly data from the National Climatic Data Center (NCDC) of the University of Wyoming website

114 (http://weather.uwyo.edu/surface/, last access: 7 April 2023). Regarding boundary layer height, daily sounding vertical

115 profiles were extracted from the national benchmark climate Nanjing station 58238 (32.00 °N, 118.48 °E) and were also

116 acquired from this website. The quality assurance and quality control (QA/QC) procedures were used at each site according

117 to the method of Xie et al. (2016) and Gao et al. (2021). $PM_{2.5}$ component data were collected hourly, and the study was

118 based on high-time resolution data. We measured 10% of all samples as parallel sampling and the pass rate was over 95%.

119 We defined the missing sampling of atmospheric pollutant data as -999 to facilitate PMF processing. The chemical mass

120 reconstruction method was used to correct potential measurement errors, which was described in detail in Section 2.2. The

121 QA/QC procedures have passed the artificial random inspection of extreme value and time consistency.

122 **2.2 Mass and chemical composition determination for $PM_{2.5}$**

123 Due to the limitation in sampling location and equipment, the sum of measured species was often lower than the

124 gravimetric mass. Chemical mass reconstruction (CMR) attempted to achieve closure between the gravitational mass and the

125 sum of components and correct potential measurement errors. In this study, the reconstructed result and the gravimetric

126 result exhibited a significant correlation, with a mean $R^2$ of 0.93, indicating that the chemical reconstruction method had

127 strong reliability. Following the work of Xu et al. (2021), eight categories of chemical components in chemically

128 reconstructed $PM_{2.5}$ can be expressed as follows:

129 $$PM_{2.5} = OM + EC + MD + TM + SO_4^{2-} + NO_3^- + NH_4^+ + Cl^- \qquad (1)$$

130 where OM refers to the organic matter. The OC to OM conversion coefficient at urban sites is 1.6 (Brokamp et al., 2017).

131 The calculation of mineral dust (MD) is based on crustal element oxides (Yan et al., 2020):

132 $$MD = 2.14 \times Si + 1.67 \times Ti + 1.89 \times Al + 1.40 \times Ca + 1.58 \times Mn + 1.43 \times Fe + 1.21 \times K + 1.67 \times Mg \qquad (2)$$

where Si is estimated as multiplying Al in crustal material by a converting factor (3.14) (Zheng et al., 2019). Trace metals
(TM) represent the sum of 30 different types of heavy metals:

$$TM = As + Co + Mo + Ag + Sc + Tl + Pd + Br + Te + Ga + Cs + Pb$$
$$+ Se + Hg + Cr + Cd + Zn + Cu + Ni + Sb + Sn + V + Ba \tag{3}$$

**2.3 Identification of source by the positive matrix factorization (PMF) model**

The positive matrix factorization (PMF) was developed by the Environmental Protection Agency (EPA) and has been
widely adopted to classify PM$_{2.5}$ into different factors (Zong et al., 2016). The US EPA PMF version 5.0 was referred to in
this study. The basic principle of the PMF model was to calculate the weight error of each chemical component in the
particulate matter and then determined its main pollution source and contribution rate by the least square method (Paatero
and Tapper, 1994). The equation of the PMF model can be expressed as follow:

$$X_{ij} = \sum_{k=1}^{p} g_{ik} f_{kj} + e_{ij} \tag{4}$$

where $X_{ij}$ is the concentration of the $ij$th sample; $g_{ik}$ represents the contribution of the $ik$th sample; $f_{kj}$ represents the
mass fraction of the $jk$th and $e_{ij}$ is the residual between the measured mass concentration of the $ij$th sample and its analytical
value. The purpose of the PMF model is to find the minimum Q value with the concentration file and uncertainty file ($u_{ij}$)
introduced into the model. The objective function Q is defined as follows:

$$Q_{ij} = \sum_{i=1}^{n} \sum_{j=1}^{m} \left[ \frac{X_{ij} - \sum_{k=1}^{p} g_{ik} f_{kj}}{u_{ij}} \right]^2 \tag{5}$$

where Q is the sum of all sample residuals and their uncertainties u. In this study, the fitting species included 41 types of
chemical species of PM$_{2.5}$ that were selected and validated to ensure that the value of the objective function Q was
minimized.

$$Unc = \frac{5}{6} \times MDL \tag{6}$$

$$Unc = \sqrt{(Error\ Fraction \times concentration)^2 + (0.5 \times MDL)^2} \tag{7}$$

where Unc is the uncertainty. MDL is the method detection limit. If the concentration is less than or equal to the MDL
provided, Unc is calculated using a fixed fraction of the MDL (Taylor et al., 2020). If the concentration is greater than the
MDL, the calculation is based on the concentration fraction and MDL.
First, we excluded more than 50% of the dataset for species below the method detection limit (MDL) and retained 23
species that were significantly correlated with PM$_{2.5}$. Second, we calculated the uncertainty (Unc) for each species based on
the concentration fraction and MDL (Taylor et al., 2020). Third, different numbers of factors were tested with random seeds
in 20 iterations of each run. When the number of factors was set to six, the fitting degree of the model calculation results was
the highest, with a correlation coefficient of 0.93, and the species almost showed a normal curve. Finally, the bootstrap (BS)
and displacement (DISP), and BS-DISP diagnostic analysis were also used to evaluate the rationality of the apportioned
factor profiles and contributions. BS is used to detect and estimate the disproportionate effects of a small set of observations
on the solution and also, to a lesser extent, the effects of rotational ambiguity. The value of the F-peak strength was ensured
to be 0.5 to eliminate the rotation ambiguity. The mapping for each factor in this study was more than 80% from the BS run,
indicating the six-factor solution was appropriate.
**2.4 Source apportionment by backward trajectory calculation and CWT analysis**
The Hybrid Single Particle Lagrangian Integrated Trajectory (HYSPLIT) model was developed by the National
Oceanic and Atmospheric Administration (NOAA) and the Bureau of Meteorology Australia to simulate and analyze the
movement, deposition, and diffusion of airflow. The reanalysis data with a spatial resolution of one degree and a temporal
resolution of 6 h (00:00, 06:00, 12:00, and 18:00 UTC) were obtained from the Global Data Assimilation System (GDAS)
(https://rda.ucar.edu/datasets/, last access: 7 April 2023). To locate the potential source areas for the corresponding
components, we used the HYSPLIT model to analyze the backward trajectory of airflow from March 2018 to February 2019.
48-hour backward trajectories terminated at a height of 100 m above ground level were calculated at the starting point
(32.07 °N, 118.78 °E). Due to the high uncertainty of a single backward trajectory, we drew multiple trajectories and
performed cluster analysis. The cluster analysis was a multivariate statistical technique using the Angle Distance algorithm,
which could quantify the relationship among the pollution concentrations in each source area (Shu et al., 2017).
The concentration-weighted trajectory (CWT) analysis was further used to determine the relative contribution of
different areas. The CWT analysis was conducted by the TrajStat software, which was a GIS (geographic information system)
application that enabled the user to visualize and analyze the spatial and meteorological data with multiple data formats
(Feng et al., 2021). In this study, the meteorological data used for the HYSPLIT model and the CWT method remained the
same. The CWT method divided the research area into small equal grids, set a standard value for the research object, and
defined the trajectory exceeding the standard value as the pollution trajectory. According to the criteria of the Chinese
National Ambient Air Quality Standards (NAAQS), the standard value of the PM$_{2.5}$ concentrations was 75 μg·m$^{-3}$ in this
study. The spatial resolution was 0.5×0.5 (Liu et al., 2018). The CWT method reflected the pollution degree of different
trajectories by calculating the weight concentration of the airflow trajectory in potential source areas:
$$C_{ij} = \frac{1}{\sum\limits_{i=1}^{M} \tau_{ijl}} \sum\limits_{l=1}^{M} C_l \tau_{ijl} \qquad\qquad (8)$$

where $C_{ij}$ is the average weight concentration of grid $ij$, $C_l$ is the pollutant concentration based on trajectory $l$ that passes

through grid $ij$, and $\tau_{ij}$ is the residence time of trajectory $l$ in grid $ij$. Similarly, to reduce the uncertainty caused by the

smaller $n_{ij}$, the CWT value is multiplied by the weight function as well (Wong et al., 2022):

$$W_{ij} = \begin{cases} 1.00 & \left(80 < n_{ij}\right) \\ 0.72 & \left(20 < n_{ij} \leq 80\right) \\ 0.42 & \left(10 < n_{ij} \leq 20\right) \\ 0.05 & \left(n_{ij} \leq 100\right) \end{cases} \tag{9}$$

where $n_{ij}$ is the number of trajectories that pass through the $ij^{th}$ cell. $W_{ij}$ is an empirical weight function to reduce the undue

influence of small $n_{ij}$ on the CWT values (Fan et al., 2019). In this study, the CWT value of each identified source derived

from the PMF model was calculated.

**2.5 Health risk assessment**

The human health risk from heavy metals in PM$_{2.5}$ may occur through exposure to ambient air (Zhang et al., 2019).

Based on the PMF analysis, we selected six toxic elements (Cr, As, Ni, Mn, V, and Pb) for the exposure risk assessment. Cr,

Ni and As have both carcinogenic and non-carcinogenic effects, Mn and V mainly have non-carcinogenic effects, and Pb

mainly produces a carcinogenic effect (Jiang et al., 2018). The non-carcinogenic and carcinogenic risks from the toxic

species of PM$_{2.5}$ were evaluated by the hazard quotient (HQ) and lifetime carcinogenic risk (LCR), respectively. The US

EPA human health risk assessment models were used to conduct carcinogenic and non-carcinogenic risk assessments (Khan

et al., 2016):

$$EC_{inh} = \frac{GA \times ET \times EF \times ED}{AT} \tag{10}$$

$$HQ = \frac{EC_{inh}}{RfC_i \times 1000\,\mu g \cdot mg^{-1}} \tag{11}$$

$$LCR = IUR \times EC_{inh} \tag{12}$$

where $EC_{inh}$ is the average daily exposure concentration of toxic elements inhaled through respiration. $GA$ is the

concentration of toxic elements in each source composition. ET is the exposure time, 24 h·d$^{-1}$; EF is the exposure frequency,

365 d·yr$^{-1}$; ED is the exposure duration, 30 yr; and AT is the average exposure time, calculated by ED yr × 365 d·yr$^{-1}$ × 24

h·d$^{-1}$ for non-carcinogens and 70 yr × 365 d·yr$^{-1}$ × 24 h·d$^{-1}$ for carcinogens. $RfC_i$ is the inhalation reference concentration

(mg·m$^{-3}$). $IUR$ is the inhalation unit risk((μg·m$^{-3}$)$^{-1}$). HQ greater than 1 indicated a non-carcinogenic risk to human health.

For carcinogenic risk, LCR < $10^{-6}$ means no cancer risk, LCR between $10^{-6}$ and $10^{-4}$ is acceptable or tolerable, and LCR > $10^{-4}$
is intolerable. The exposure parameters were shown in Table 1 (Jiang et al., 2018; Zhang et al., 2019).

**Table 1. Exposure parameters of toxic elements through inhalation route in health risk assessments.**

| Toxic elements | RfC$_i$ (µg·m$^{-3}$)$^{-1}$ | IUR (mg·m$^{-3}$) |
|---|---|---|
| Cr | $1.0\times10^{-4}$ | $1.2\times10^{-2}$ |
| As | $1.5\times10^{-5}$ | $4.3\times10^{-3}$ |
| Ni | $1.4\times10^{-5}$ | $2.4\times10^{-4}$ |
| Mn | $5.0\times10^{-5}$ | —— |
| V | $1.0\times10^{-4}$ | —— |
| Pb | —— | $1.2\times10^{-5}$ |


## 3. Results and discussions
### 3.1 Chemical components, meteorological parameters and diurnal variations
Table 2 shows the seasonal average of chemical components and meteorological parameters from March 2018 to
February 2019. In this study, March to May 2018 is defined as spring, June to August 2018 is defined as summer, September
to November 2018 is defined as fall, and December 2018 to February 2019 is defined as winter. The daily average
concentration of PM$_{2.5}$ ranged from 6.7 to 234.0 µg·m$^{-3}$, with an annual average of 68.7 µg·m$^{-3}$. The order of average
concentrations of PM$_{2.5}$ in each season was winter (113.9 µg·m$^{-3}$) > spring (99.1 µg·m$^{-3}$) > autumn (38.9 µg·m$^{-3}$) > summer
(23.7 µg·m$^{-3}$). Seasonal variations of PM$_{2.5}$ were closely related to emission and meteorological conditions. In spring, the
wind speed (WS) was higher (3.5 m·s$^{-1}$) than those in other seasons. Pearson correlation showed that PM$_{2.5}$ concentrations
were significantly (p<0.01) correlated to WS (r=-0.36) in spring. In summer, high boundary layer height (BLH) (520.4m)
significantly reduced PM$_{2.5}$ concentrations. In autumn and winter, PM$_{2.5}$ showed significant correlations between
temperature (r=-0.53), relative humidity (r=0.62) and BLH (r=-0.43). Biomass burning and industrial emissions are
important sources of aerosols in the urban atmosphere and contribute 7-27% to PM$_{2.5}$ mass in applicable cities (Tao et al.,
2017; Andreae et al., 2019). Coal consumption and population density have a significantly positive effect on PM$_{2.5}$
concentration (Zhou et al., 2018; Chow et al., 2022). The highest level of PM$_{2.5}$ in winter was due to coal consumption,
lower temperatures (4.9°C), higher humidity (79.6%), and lower BLH (419.7m) than in summer.
The seasonal variation of anthropogenic emissions also considerably affected PM$_{2.5}$ concentrations. The order of the
major components in PM$_{2.5}$ was NO$_3^-$ (20-31%) > SO$_4^{2-}$ (16-27%) > NH$_4^+$ (11-19%) > mineral dust (8-14%) > OM (6-
14%) > EC (2-4%) > trace metals (2-3%) > Cl$^-$ (1-3%). Sulfate, nitrate, and ammonium (SNA) accounted for 60% of the
total PM$_{2.5}$ and were closely related to the secondary transformation of gaseous precursors. The concentration ratio of NO$_3^-$
to SO$_4^{2-}$ (NO$_3^-$/SO$_4^{2-}$) was used to differentiate the relative importance of nitrogen (generally related to vehicle emissions)
and sulfur (normally related to stationary sources) in the atmosphere (Liu et al., 2019). Over the past few years, the mass

ratio of $NO_3^-/SO_4^{2-}$ was 2.13 in Ningbo, 1.89 in Hangzhou, and 1.21 in Beijing (Huang et al., 2017; Li et al., 2018). In this study, the average ratios of $NO_3^-/SO_4^{2-}$ were 1.81 in spring, 1.20 in summer, 2.34 in autumn, and 1.59 in winter, respectively, indicating the enhanced secondary transformation of gaseous pollutants (e.g. $SO_2$, NOx, VOCs) during heavily polluted periods (Liu et al., 2016; Liu et al., 2018). The oxidation rates of $SO_2$ and $NO_2$ need to be further investigated. Carbonaceous aerosols (OM and EC) accounted for 12% and 14% of $PM_{2.5}$ in spring and winter, respectively. The large increase in the number of coal fires used for residential heating in winter may increase the abundance of carbon-containing emissions, including OC, EC, and VOCs (Islam et al., 2020). Compared with 2015, the concentrations of OM and EC decreased from 22.9% to 12.8% (Chen et al., 2017). This may be related to policies to control coal combustion and motor vehicle emissions, considering similar meteorological conditions in the two periods (Tao et al., 2017; Jeong et al., 2019).

Table 2. Seasonal average concentration of components of $PM_{2.5}$, in $\mu g \cdot m^{-3}$ and % in brackets, and meteorological parameters. T, RH, WS, and BLH represent air temperature, relative humidity, wind speed and boundary layer height, respectively.

| Components and meteorological parameters | Spring | Summer | Autumn | Winter |
|---|---|---|---|---|
| $PM_{2.5}$ | $99.1 \pm 29.5$ | $23.7 \pm 12.2$ | $38.9 \pm 20.6$ | $113.9 \pm 43.6$ |
| $SO_4^{2-}$ | $20.5 \pm 5.9\ (20.7)$ | $5.2 \pm 2.1\ (21.9)$ | $7.3 \pm 4.8\ (18.8)$ | $31.5 \pm 8.7\ (27.7)$ |
| $NO_3^-$ | $16.9 \pm 11.4\ (17.1)$ | $5.3 \pm 1.2\ (22.4)$ | $9.8 \pm 3.3\ (25.2)$ | $27.2 \pm 17.5\ (23.9)$ |
| $NH_4^+$ | $15.1 \pm 6.1\ (15.2)$ | $3.2 \pm 1.7\ (13.5)$ | $7.1 \pm 2.1\ (18.3)$ | $11.5 \pm 4.6\ (10.1)$ |
| OM | $11.7 \pm 6.1\ (11.8)$ | $1.6 \pm 0.7\ (6.8)$ | $4.1 \pm 1.1\ (10.5)$ | $11.0 \pm 5.8\ (9.7)$ |
| EC | $2.3 \pm 0.8\ (2.3)$ | $0.8 \pm 0.3\ (3.4)$ | $1.6 \pm 1.2\ (4.1)$ | $3.6 \pm 1.5\ (3.2)$ |
| Mineral dust | $13.2 \pm 4.5\ (13.3)$ | $2.3 \pm 0.8\ (9.7)$ | $2.7 \pm 1.0\ (6.9)$ | $8.7 \pm 2.7\ (7.6)$ |
| Trace metals | $2.7 \pm 1.5\ (2.7)$ | $0.5 \pm 0.1\ (2.1)$ | $0.5 \pm 0.2\ (1.3)$ | $1.6 \pm 0.9\ (1.4)$ |
| $Cl^-$ | $2.7 \pm 0.9\ (2.7)$ | $1.6 \pm 0.6\ (6.8)$ | $0.8 \pm 0.2\ (2.1)$ | $1.7 \pm 0.4\ (1.5)$ |
| T (°C) | $18.8 \pm 4.3$ | $27.6 \pm 5.4$ | $19.4 \pm 4.9$ | $4.9 \pm 2.2$ |
| RH (%) | $86.5 \pm 12.9$ | $58.2 \pm 6.3$ | $73.1 \pm 8.5$ | $79.6 \pm 10.4$ |
| WS ($m \cdot s^{-1}$) | $3.5 \pm 0.6$ | $2.9 \pm 0.5$ | $2.7 \pm 0.5$ | $2.1 \pm 0.3$ |
| BLH (m) | $469.7 \pm 40.9$ | $520.4 \pm 58.9$ | $443.6 \pm 32.4$ | $419.7 \pm 23.5$ |

Figure 1 shows the diurnal variation of chemical components in $PM_{2.5}$. The seasonal differences were mainly reflected in the variation in the timing of peak values. In spring (Fig. 1a), the highest and lowest $PM_{2.5}$ concentrations were 143.6 $\mu g \cdot m^{-3}$ at 7:00 and 94.8 $\mu g \cdot m^{-3}$ at 14:00, respectively. The concentration of SNA had obvious diurnal variations. From 6:00 to 18:00, the average concentration of $NO_3^-$ increased from 17.6 to 21.8 $\mu g \cdot m^{-3}$, while the average concentration of $SO_4^{2-}$ decreased from 23.2 to 15.9 $\mu g \cdot m^{-3}$. In summer (Fig. 1b), the highest and lowest $PM_{2.5}$ concentrations were 23.5 $\mu g \cdot m^{-3}$ at 9:00 and 14.2 $\mu g \cdot m^{-3}$ at 14:00, respectively. The maximum concentration difference of SNA between day and night was less

than 10 µg·m⁻³, indicating the study area was in a relatively stable background field (Chen et al., 2018). In autumn (Fig. 1c), the highest and lowest PM$_{2.5}$ concentrations were 77.1 µg·m⁻³ at 8:00 and 47.8 µg·m⁻³ at 16:00, respectively. The concentration of SNA increased at night and decreased during the day. The maximum concentration difference was more than 20 µg·m⁻³. In winter (Fig. 1d), from 18:00 to 23:00, the concentration of SNA increased from 74.5 µg·m⁻³ to 108.7 µg·m⁻³, with increasing rates of 8.5 µg·m⁻³·h⁻¹. The height of the atmospheric boundary layer decreased early in the winter afternoons (Chen et al., 2018). The values of PM$_{2.5}$ in winter were higher at night due to the coal combustion and biomass burning (BB) for residential heating (Zou et al., 2017). In summary, compared with the spring and winter, PM$_{2.5}$ presented similar and relatively flat diurnal patterns in both autumn and summer. Although the seasonal variations of mass concentrations and aerosol compositions were substantially different, the concentrations of aerosol species showed similar diurnal variation patterns during all the sampling days with higher values in the nighttime and early morning, suggesting that the factors driving the diurnal variations were similar.

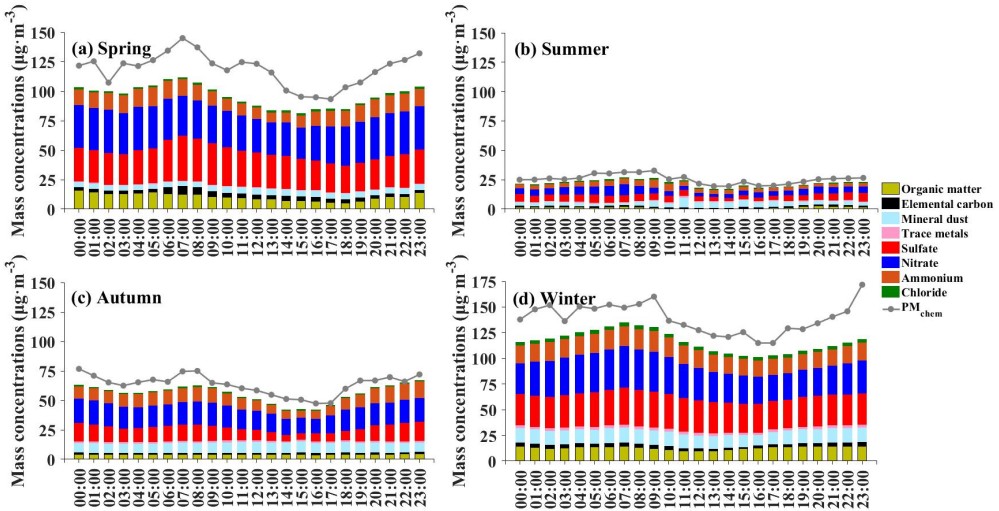

**Figure 1. Average diurnal variation of the concentrations of major chemical components of PM$_{2.5}$ per each season.**

**3.2 Variation of PM$_{2.5}$ chemical compositions at different pollution levels**

Figure 2 presents the PM$_{2.5}$ concentrations and components at different pollution levels. In this study, it was defined as the clean day (C) when the daily average PM$_{2.5}$ concentrations were less than 35 µg·m⁻³, the moderate pollution day (MP) when PM$_{2.5}$ concentrations were more than 35 µg·m⁻³ and less than 150 µg·m⁻³, and the heavy pollution day (HP) when PM$_{2.5}$ concentrations were more than or equal to 150 µg·m⁻³. As shown in Figure 2a, the annual average concentration of the water-soluble inorganic ions (WSIIs) was 41.9 µg·m⁻³, and accounted for 61.8% of PM$_{2.5}$. WSIIs were largely responsible for the variability in PM$_{2.5}$. The ratios of SNA in spring and winter were similar, with ratios of 65.0% for clean days, 75.0% for moderate pollution days, and 83.9% for heavy pollution days. With the degradation of air quality, the contribution of NO$_3^-$

noticeably increased from 27.5% to 47.8% in spring and from 28.9% to 44.7% in autumn. To understand the oxidation rates
of $SO_2$ and $NO_2$, the sulfur oxidation rate and nitrogen oxidation rate (defined as SOR = $SO_4^{2-}/(SO_4^{2-}+SO_2)$ and NOR =
$NO_3^-/(NO_3^- + NO_2)$) were calculated. The critical value of SOR and NOR in the atmosphere are both 0.1 (Win et al., 2020).
The order of the seasonal average NOR was winter (0.21) > spring (0.18) > autumn (0.17) > summer (0.15), while the order
of the seasonal average SOR was winter (0.51) > spring (0.43) > autumn (0.42) > summer (0.36). $PM_{2.5}$ pollution in winter is
associated with high RH and rapid production of particulate sulfate from the oxidation of $SO_2$ emitted by coal combustion
(Wang et al., 2020). From summer to winter, the NOR and SOR values increased by 40.0% and 41.6%, respectively. SOR
and NOR showed a strong positive correlation with relative humidity, with a correlation coefficient of 0.53 and 0.61,
respectively. The contribution of coal combustion varied between 30 and 57% of $PM_{2.5}$ in winter (Zhang et al., 2017). Under
the conditions of high coal combustion emissions and high RH, the rapid oxidation of $SO_2$ occurred to produce sulfate. The
sensitivity of $PM_{2.5}$ to surface temperature, wind speed, and boundary layer height is negative, while the sensitivity to
relative humidity is positive (Chen et al., 2018; Sulaymon et al., 2021). In summer, the correlation coefficients of $PM_{2.5}$ with
RH, T, WS, and BLH were 0.42, -0.47, -0.15, and -0.23, respectively. In winter, the correlation coefficients of $PM_{2.5}$
concentration with RH, T, WS, and BLH were 0.74, -0.57, -0.31, and -0.32, respectively. High RH (79.6%), low temperature
(4.9°C), low WS (2.1m·s$^{-1}$), and low BLH (419.7m) provided favorable conditions for the accumulation of $PM_{2.5}$.
Coal combustion, biomass burning, and motor vehicle emissions all lead to a remarkable increase in carbonaceous
aerosols (Chow et al., 2022). As shown in Figure 2b, carbonaceous species also had a significantly enhanced contribution in
the colder season compared to the warmer season. The seasonal differences might be related to the effects of meteorological
conditions and source emissions. Pearson correlation showed that the relationships between OM and EC and meteorological
parameters (T, RH, WS, and BLH) were not significant (Table 2). To explore the possible pollution sources, it is feasible to
study the mass ratio of OC/EC under different pollution levels. OC comprises thousands of organic compounds. EC is stable
and mainly derived from primary sources of combustion products (Zhang et al., 2017; Wu et al., 2020). The OC/EC mass
ratio of motor vehicle emissions (1.1) is lower than that of coal combustion (2.7) and biomass burning (9.0) (Xu et al., 2021).
In this study, the OC/EC ratios continuously decreased as air pollution got worse, and the values ranged from 6.1 (C), 4.1
(MP) to 3.9 (HP) in spring, from 6.2 (C) to 4.8 (MP) in autumn and from 4.3 (C), 2.7 (MP) to 1.3 (HP) in winter. The annual
average ratio of OC/EC decreased by 56.1% from clean days to heavy pollution days.  If the OC/EC values were in the range
of 2.5-5.0, vehicle exhaust emissions were considered as the main source of OC and EC in aerosols, whereas if the OC/EC
values were in the range of 5.0-10.5, coal combustion was considered the main source of OC and EC in aerosols (Gao et al.,
2018; Liu et al., 2018). Distinct differences in the evolution of the OC/EC ratio on polluted days imply that mobile sources
are likely more important. Both the increase in motor vehicle emissions and the formation of meteorological conditions
conducive to pollutant accumulation contribute to the decrease in the OC/EC ratio

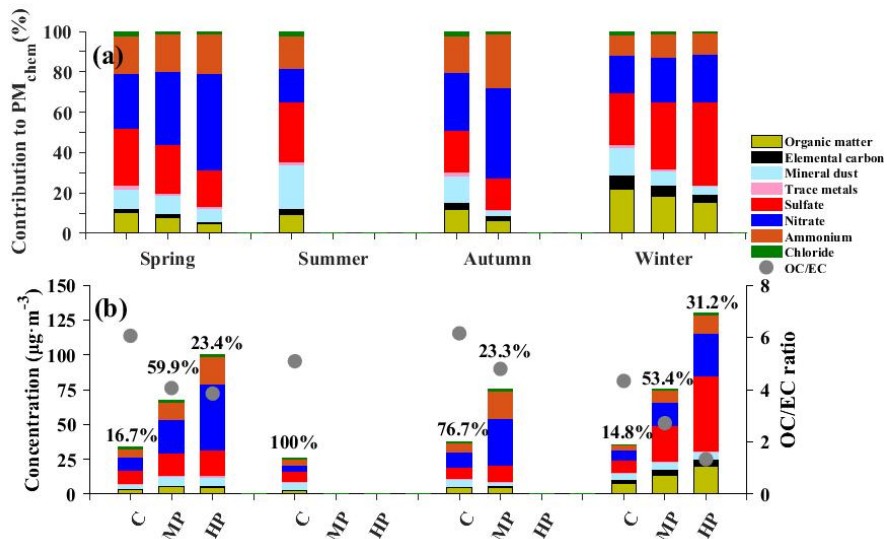

Figure 2. Chemical compositions of PM$_{2.5}$ and mass ratio of OC/EC at different pollution levels of the total samples per season. C, MP, and HP represent the clean day, moderately polluted day, and heavily polluted day, respectively. "%" represents the proportion of the filter sample quantity at each pollution level out of the total samples.

**3.3 Source identification and apportionment**

**3.3.1 Elemental profile and source apportionment from the PMF model**

To further quantitatively determine the source apportionment of PM$_{2.5}$, the EPA PMF5.0 model was adopted. The number of factors in the PMF model corresponded to the number of sources of PM$_{2.5}$ in this study. When the number of factors was set to six, the fitting degree of the model calculation results was the highest. Figure 3 presents the factor profiles and relative contributions of six factors to each species (% of species total), including secondary inorganic aerosol source (SIS), coal combustion (CC), industry source (IS), vehicle emission (VE), fugitive dust (FD), and other sources (OS). The meaning of % is the proportion of each chemical component in each source of PM$_{2.5}$. As shown in Figure 3a, the compositions of SIS were more clear than other sources. NO$_3^-$ and SO$_4^{2-}$ are mainly from the oxidation of NO$_X$ and SO$_2$, while NH$_4^+$ probably comes from the conversion processes between ammonia and sulfuric and nitric acid (Win et al., 2020). Factor 1 was identified as the SIS with distinctly high loads of NH$_4^+$ (66.9%), NO$_3^-$ (61.9%), SO$_4^{2-}$ (63.8%) and Cl$^-$ (55.3%). In Figure 3b, the high proportion of Pb (38.2%) and Se (45.1%) was identified in Factor 2, associating with moderate weighting on As (14.3%), SO$_4^{2-}$ (20.5%), and Cl$^-$ (22.2%). Pb and As are important identifying elements of coal combustion and are used as tracers (Xie et al., 2020). SO$_4^{2-}$ is formed by the photochemical oxidation of sulfur-containing precursors (SO$_2$ and H$_2$S) released by coal combustion (Zong et al., 2016). Given the source profile, Factor 2 was related to coal combustion emissions. Factor 3 (Figure 3c) was characterized by the association of heavy metal pollutants such as As (42.8%), Pb (33.8%), Cr (61.1%), Zn (58.9%), Cu (59.4%), Fe (38.3%), and Mn (40.1%). As, Pb, Cr, Fe and Mn are related to metal smelting and processing (Fang et al., 2021). However, the percentage of OC was only 11.3%, while rates of Zn

(58.9%) and Cu (59.4%) were higher in Factor 3 (Fig. 3c). Cu, Zn, and OC are used as tracers of a mixed source of traffic
and industrial, and OC is the major pollutant in the vehicle exhaust (Wang et al., 2020). Compared to motor vehicle
emissions, Factor 3 should be significantly influenced by industrial activities. Cu and Zn were mainly from industrial process
sources. As discussed above, Factor 3 was attributed to the IS. Factor 4 (Figure 3d) was characterized by the association of
vehicle emissions, with the high proportions of Ni (54.7%), V (80.5%), OC (55.4%), EC (79.8%), and $NO_3^-$ (20.3%). VOCs
and NOx released from vehicles were the precursors of the secondary organic compounds and nitrate in $PM_{2.5}$ and were
important catalysts for increased atmospheric oxidation (Guevara et al., 2021). OC and EC are mainly from the vehicle
exhaust, and Ni and V are usually tracers of heavy oil combustion (Wu et al., 2020; Veld., 2021). Factor 4 contained a high
proportion of OC, EC, and $NO_3^-$, which could be considered as vehicle emission, while factor 4 contained Ni and V, which
were also influenced by shipping emissions (Gao et al., 2018; Veld., 2021). As shown in Figure 3e, Factor 5 had relatively
high proportions of Fe (31.1%), Ti (78.2%), $K^+$ (55.8%), $Ca^{2+}$ (60.5%), and $Mg^{2+}$ (48.3%). Ti, Fe, and Mg are both common
crustal elements that can represent the source of mineral dust. $K^+$ and $Ca^{2+}$ are considered to be significant tracers of biomass
burning, which have obvious seasonal variations (Tong et al., 2020; Silva et al., 2022). Factor 5 was classified as the fugitive
dust and biomass burning, including road dust, industrial dust, and soil dust. Factor 6 (Fig. 3f) was unidentified and could be
affected by coal combustion, industrial processes, and biomass burning. In the absence of a clear designation of the source,
Factor 6 was attributed to an erroneous contribution from a different source.

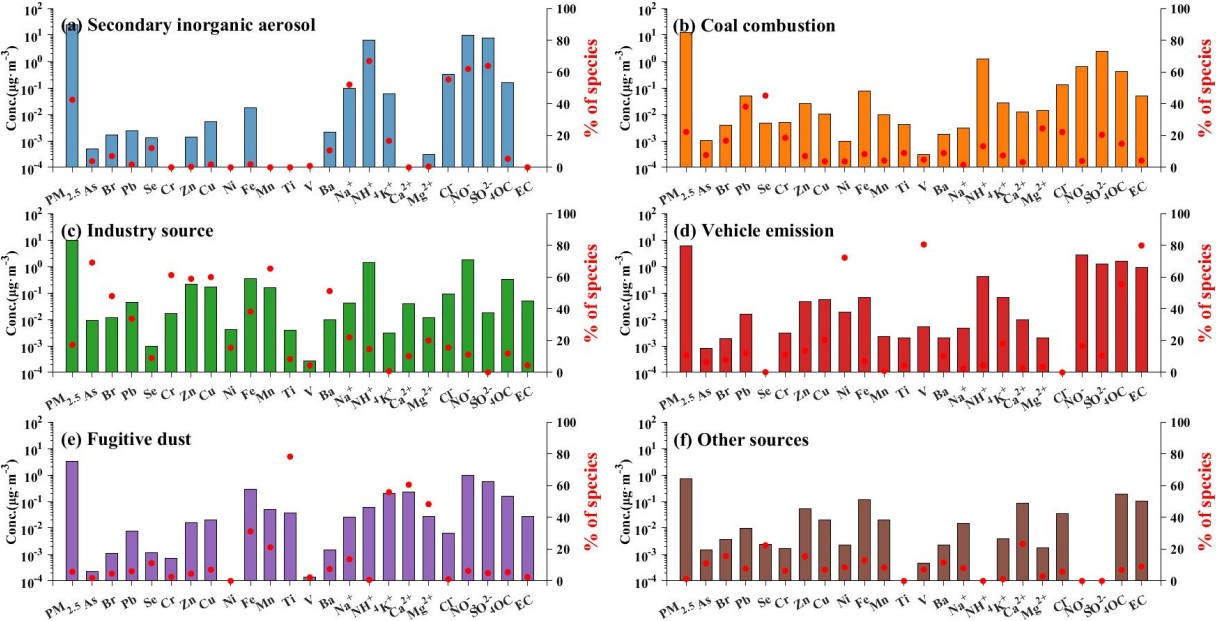

**Figure 3. Factor profile in each source for PM₂.₅ during the whole year. The histograms are the mass concentration of each species**
**to every species (μg·m⁻³), and the red dots are the relative contributions of each source to every species (%).**

Figure 4 and Figure 5 show the comparisons of our PMF results with the previous findings. In the YED region, SIS
contributed about 42.5% to PM$_{2.5}$ in Nanjing in this study, which was higher than that reported by Li et al. (2020), while the
contributions of CC was lower. However, other sources of PM$_{2.5}$ in different cities were more complicated. In the BTH, IS
was crucial source and contributed about 30% of Tianjin and Shijiazhuang (Huang et al., 2017). In contrast, IS in Nanjing
contributed only 17.3% of PM$_{2.5}$ pollution. Recent emission control policies in the YRD have had positive effects on
reducing industrial pollution. In the PRD, vehicle emissions, secondary nitrate, coal burning, and industrial emissions
showed obvious local emission characteristics. An extra 30% PM$_{2.5}$ concentration was tightly related to local emissions in
the downtown and industrial areas (Huang et al., 2014; Li et al., 2020; Chow et al., 2022). In this study, VE contributed only
10.7% in Nanjing. It is worth noting that the PMF model assumes that source profiles do not change significantly over time
and that species do not undergo chemical reactions (Paatero and Tapper, 1994). The human activities under seasonal
variations in this study made the actual pollution incompatible with the ideal assumption. For example, emissions from coal
combustion increased the contribution of CC in winter significantly (Xu et al., 2021). In addition, the sources of air masses
in each season also created uncertainties. All of these required detailed discussions of regional transport conditions in each
season.

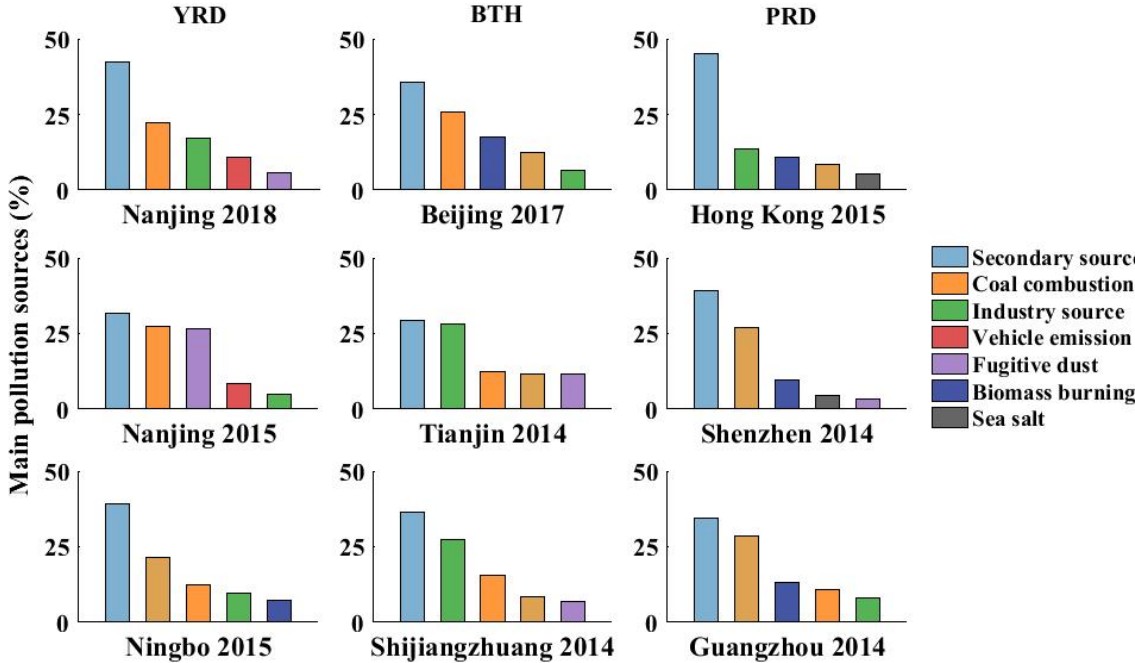

**Figure 4. Comparisons of source apportionment for PM$_{2.5}$ among different cities.**

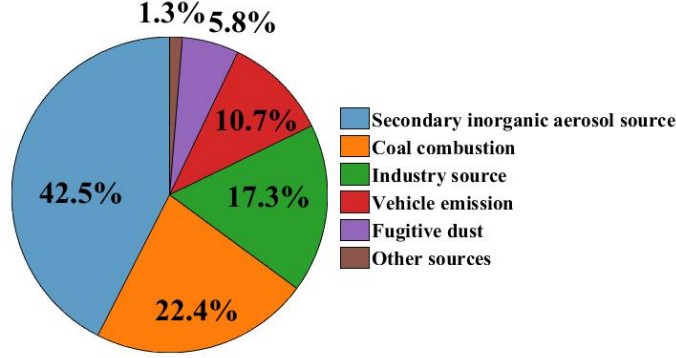

**Figure 5. Average annual contribution of the sources identified for PM$_{2.5}$ in Nanjing in 2018.**

### 3.3.2 Source identification by backward air mass trajectory analysis

The regional transport of air pollutants exerts a profound impact on local air quality (Shu et al., 2017). Figure 6 shows the quantified contributions of PM$_{2.5}$ with 48-h backward trajectories. In spring (Fig 6a), nearly half of the air masses (cluster a3) stemmed from northern Jiangxi, passed over Anhui Province before arriving at the sampling sites, and had the highest PM$_{2.5}$ average value (127.2 μg·m$^{-3}$). CC from cluster a3 had the highest contribution with mass and percentage contributions being 41.0 μg·m$^{-3}$ and 32.3%, respectively. In addition, FD contributed relatively highly in clusters a2 and a3, with proportions of 18.2% and 10.3%, respectively. Increased contribution from fugitive dust was related to industrial and construction activities (Xu et al., 2016). Cluster a2 originated in Liaoning and Cluster a3 was from northern Jiangxi. There were many industrial cities located in Liaoning, and the largest coal-fired thermal power plant in Jiangxi was located in the northern city of Jiujiang (Xu et al., 2016; Wang et al., 2019). Long-range transport of dust from these areas would have a high impact on the formation of severe particle pollution in the YRD. In summer (Fig 6b), the most obvious characteristic of regional transport was significantly influenced by the ocean. Clusters b2 and b3 were relatively clean with low concentrations of PM$_{2.5}$ (28.2 μg·m$^{-3}$ for b2 and 32.4 μg·m$^{-3}$ for b3). These clusters passed over the ocean areas and accounted for more than half of all trajectories. The magnitude of total CC, IS and VE exhibited a descending order from clusters b1 to b3. The dilution effects of clean ocean air masses played a vital role in particulate pollution. In autumn (Fig 6c), there were the highest concentrations of PM$_{2.5}$ in cluster c1, with an average value of 84.6 μg·m$^{-3}$. CC (23.1%) and IS (27.6%) contributed relatively highly in cluster c1, indicating that regional transport from industrial regions might play an important role. For SIS, the proportion of NH$_4^+$ in these air masses was significantly higher in autumn than those in other seasons (Table 2). The increase in the proportion of NH$_4^+$ indicated that air pollution masses were heavily affected by nearby agricultural activities. In winter (Fig 6d), clusters d1 (108.3 μg·m$^{-3}$) and d3 (122.6 μg·m$^{-3}$) originated from Shandong Province and the BTH, accounting for more than three-quarters of the air masses. These air masses, which moved at high altitudes with a slow speed, could have carried abundant air pollutants. Cluster d2 (153.9 μg·m$^{-3}$) was short-distance transport and derived from Jiangsu Province. The contribution of SIS exhibited a increasing order from clusters d1 (22.9%)

to d3 (43.4%) to d2 (55.5%), corresponding to the transition from long-range transport air masses to short-distance transport air masses.

Figure 7 shows the spatial distribution of the contribution from each source of PM$_{2.5}$ by the CWT method and highlighted the potential geographic origins. For SIS (Fig. 7a), the high levels (10-15 µg·m$^{-3}$) of this source mainly originated from local emissions in Jiangsu and regional transport from Shandong Province. For CC (Fig. 7b), the high emissions (10-11 µg·m$^{-3}$) were distributed in the YRD and central China. The weighted concentration values of CC were lower than those of the SIS. High concentrations near the center area are associated with local sources, while those far away from the center area are indicative of regional transport (Shu et al., 2017). The secondary aerosol source was probably from the accumulation of precursors emitted by local emissions. For IS and VE (Fig. 7c and d), there were no high potential areas for these sources. The moderate weighted concentration values of IS (8-10 µg·m$^{-3}$) were potentially located in the north of Jiangsu, Anhui, and Jiangxi, which were the most important industrial base in China. The oceanic air masses are influenced by tropical cyclones with high temperature and strong wind (Li et al., 2018; Chow et al., 2022). Based on the backward trajectory calculation, most of the long-range transport of PM$_{2.5}$ passed through the Yellow Sea and the East Sea. High wind speed had a great effect on mitigating PM$_{2.5}$ pollution.

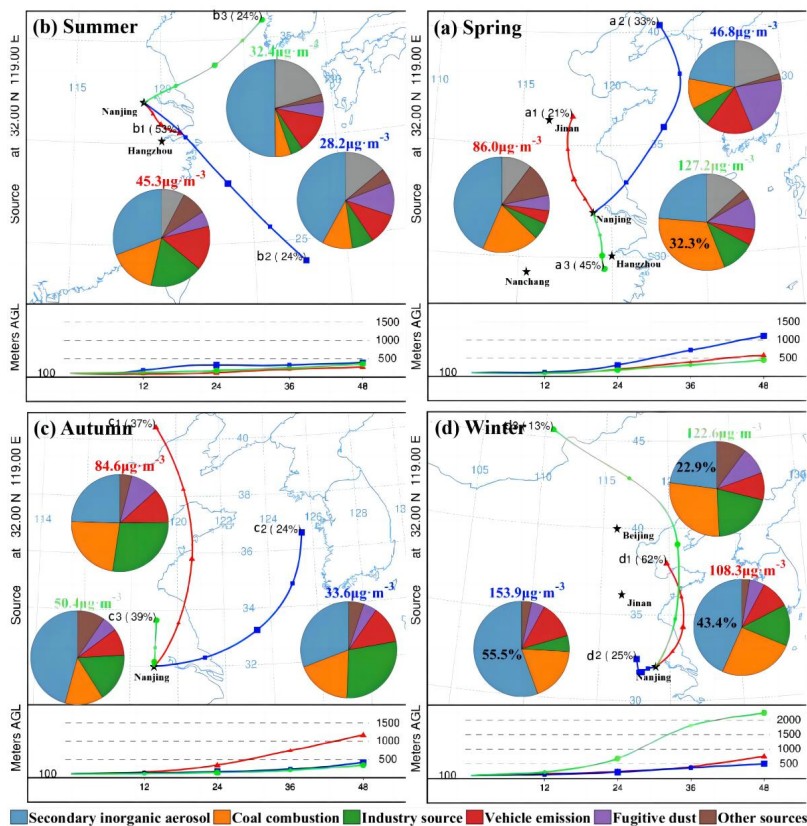

**Figure 6. Source contributions to PM$_{2.5}$ grouped by air masses associated with different 48-h backward trajectory clusters. The pie charts show the average source contribution for corresponding clusters.**

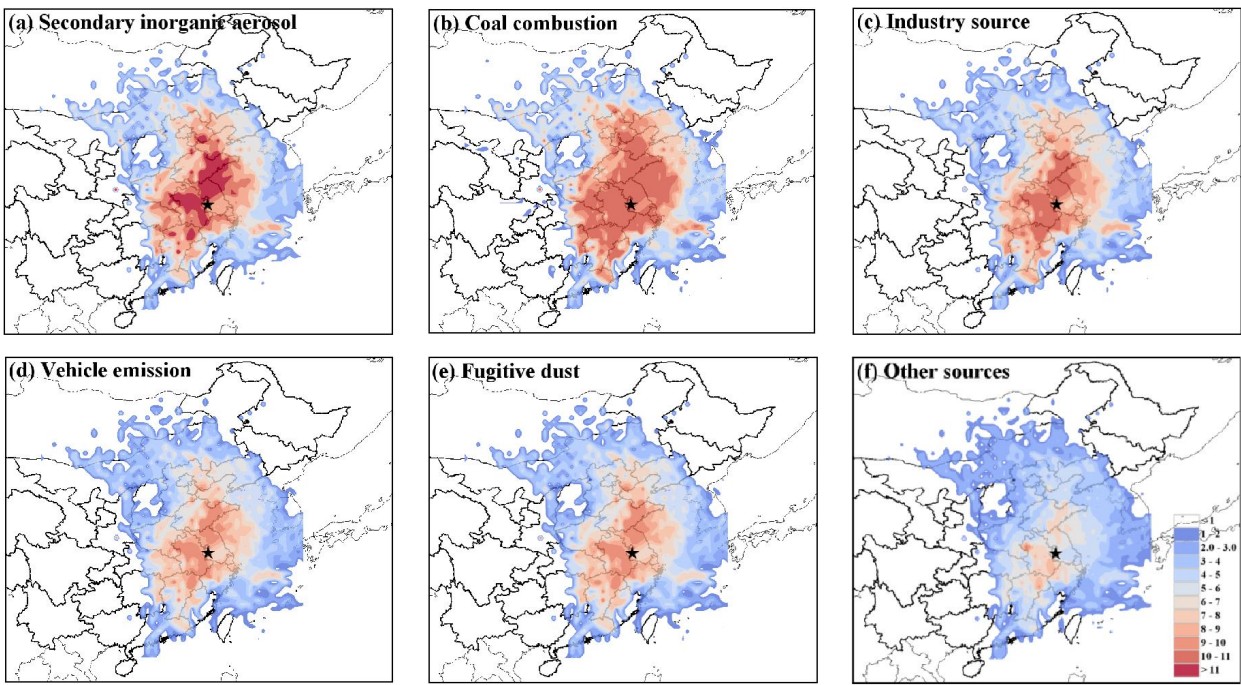

417

**Figure 7. Potential source regions for individual sources of PM$_{2.5}$ identified by the CWT method from March 2018 to February 2019.**

**3.4 Non-carcinogenic and carcinogenic health risks of toxic metal elements in each source of PM$_{2.5}$**

Figure 8 shows the HQ values of non-carcinogenic and the LCR values of carcinogenic risks in PM$_{2.5}$ and their total health risk in each source. For non-carcinogenic risk (Fig. 8a), the order of the average HQ values was Mn(0.47) > Ni (0.32) > As (0.14) > Cr (0.04) > V (0.02). The HQ values of toxic elements were all less than one, which indicated that there was no significant non-carcinogenic risk. However, the summation of five HQ values was higher than one, indicating that the combined exposure to the pollutant class still had adverse effects. The carcinogenic risk (Fig. 8b) posed by Ni ($2.3\times10^{-7}$) and Pb ($6.8\times10^{-8}$) were lower than $1\times10^{-6}$ and could be acceptable. The carcinogenic risk level of Cr ($1.0\times10^{-7}$) and As ($1.8\times10^{-5}$) were within the tolerance or acceptable level ($1\times10^{-6}$-$1\times10^{-4}$) (Zheng et al., 2019). Figures 8 c and d show the integrated assessment of the source apportionment in toxic elements. IS accounted for the largest proportion of the non-carcinogenic and carcinogenic risk, with the HQ of 0.83 and the LCR of $5.8\times10^{-6}$, respectively. Although the PMF results indicated that SIS had the highest contribution to PM$_{2.5}$ (Fig. 5), the health risk results showed that the health risks of toxic elements from IS and CC were much higher than those from SIS. Previous studies showed that coal combustion sources in Beijing, Shanxi, and Jinan had higher respiratory exposure and health risks, while the fugitive dust source in Liaoning contained higher levels of Pb, As, and Co (Zeng et al., 2019). As, Cr, and Ni in PM$_{2.5}$ were within the acceptable level for both children and adults in Nanjing, but there was a potential carcinogenic risk posed by Pb via ingestion to children and

adults (Hu et al., 2012). It was related to the differences in PM$_{2.5}$ pollution characteristics and source contributions in
different cities. The ingestion exposure may result in the potential health risk from IS, CC, and VE. Based on the
implementation of energy conservation and emission reduction policies, the main source of pollution in Nanjing is SIS at
present, and the health risk has been alleviated. However, we should pay more attention to the health burden of vehicle
emissions, coal combustion, and industrial processes.

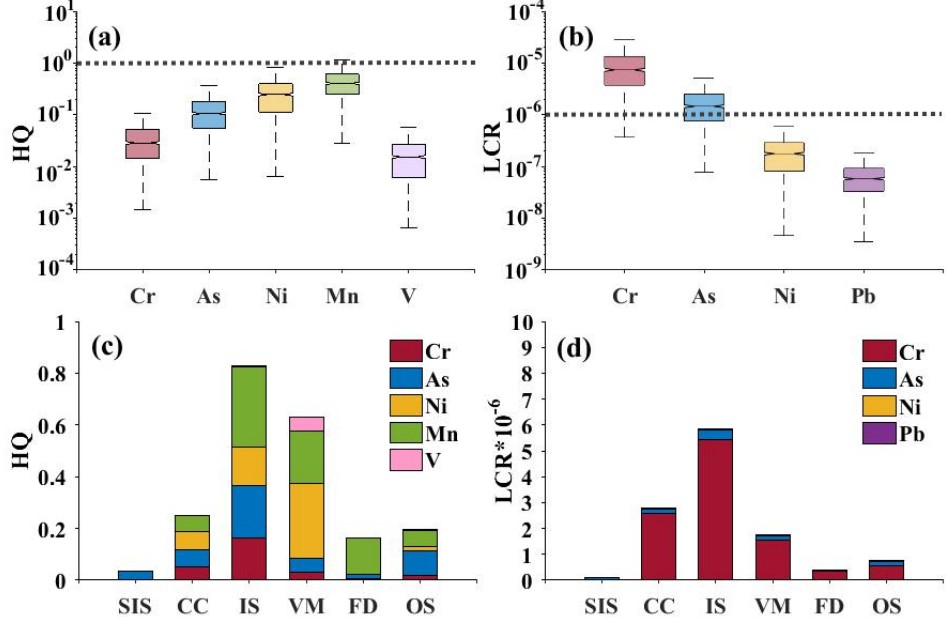

**Figure 8. Non-carcinogenic (a) and carcinogenic (b) risks of toxic elements. Non-carcinogenic (c) and carcinogenic (d) risk of the**
**sources identified for PM$_{2.5}$ in Nanjing. HQ, LCR, SIS, CC, IS, VM, FD, and OS represent hazard quotient, lifetime carcinogenic**
**risk, secondary inorganic aerosol source, coal combustion, industry source, vehicle emission, fugitive dust, and other sources,**
**respectively.**

### 3. Conclusions

Identifying and quantitatively assessing the contributions and health risks of pollution sources has played an important
role in formulating policies to control particle pollution. We have derived a high-quality PM$_{2.5}$ composition data set, based
on a chemical component monitoring from March 2018 to February 2019 in Nanjing. The PMF and back-trajectory results
were adopted to investigate the chemical characteristics and regional transports of each source. The health risk assessment
was used to explore non-carcinogenic and carcinogenic risks of toxic elements.
The results showed that PM$_{2.5}$ concentrations ranged from 6.7 to 234.0 μg·m$^{-3}$, with an annual average of 68.7 μg·m$^{-3}$.
Water-soluble ions contributed the most to PM$_{2.5}$. From summer to winter, the NO$_3^-$/SO$_4^{2-}$ ratio increased from 1.2 to 1.59.
OC/EC ratio decreased by 56.1% from clean days to heavy pollution days. The average OC/EC ratio on heavy pollution days
was 1.3. Both the increase in motor vehicle emissions and the formation of meteorological conditions conducive to pollutant
accumulation contribute to the decrease in the OC/EC ratio. Based on the PMF model, the source variations and health risks
were assessed. The contribution of identified sources (including SIS (42.5%), CC (22.4%), IS (17.3%), VE (10.7%), FD
(5.8%), and other sources (1.3%)) had different spatial distributions and seasonal variations. The CWT analysis indicated
that high emissions (10-11 $\mu g \cdot m^{-3}$) of SIS and CC were distributed in the YRD and central China in winter. Moderate
emissions (8-9 $\mu g \cdot m^{-3}$) of IS and VE were potentially located in the north of Jiangsu, Anhui, and Jiangxi. The carcinogenic
and non-carcinogenic risks of toxic elements (Cr, As, Ni, Mn, V, and Pb) mainly came from IS, VE, and CC, which were
within the tolerance or acceptable level. Based on the implementation of energy conservation and emission reduction
policies, the main source of pollution in Nanjing is SIS at present, and the health risk has been alleviated. However, we
should pay more attention to the health burden of vehicle emissions, coal combustion, and industrial processes.

467        This study provided new insight for $PM_{2.5}$ research between the source apportionment and health risk. The results

presented characteristics of chemical components, pinpointed secondary transformation processes leading to the high $PM_{2.5}$
concentrations, revealed spatial variations of source contribution, and provided new references for mega-cities to conduct
health risk analysis on air pollution control measures.

*Data Availability.*
$PM_{2.5}$ composition data were collected by the atmospheric heavy metal Monitor and the In-situ Gas and Aerosol
Compositions Monitor in the School of Atmospheric Sciences, Nanjing University (The data presented in this article are
available upon request from Yangzhihao Zhan (zyzh1049744276@gmail.com)). Air quality monitoring data were acquired
from the official NEMC real-time publishing platform (https://air.cnemc.cn:18007/, last access: 7 April 2023).
Meteorological data were obtained from the University of Wyoming website (http://weather.uwyo.edu/, last access: 7 April
2023). The NCEP FNL data were taken from the NCEP (https://rda.ucar.edu/datasets/, last access: 7 April 2023). These data
can be downloaded for free as long as one agrees to the official instructions.

*Author contributions.*
YZ and MX had original ideas, designed the research, collected the data, and prepared the original draft. YZ, WZ, PC, YL,
and RZ performed PMF experiments and carried out the data analysis. MX and WZ acquired financial support for the project
leading to this publication. TW, DG, JT, KZ, SL, BZ, and ML reviewed the initial draft and checked the English of the
original paper.

*Acknowledgments.*
The authors are grateful to NEMC for the air quality monitoring data, to NCDC for the meteorological data, and to NCEP for
global final analysis fields. We gratefully acknowledge the NOAA Air Resources Laboratory (ARL) for providing the

HYSPLIT transport and dispersion model used in this work. We acknowledge the Chinese Academy of Meteorological Sciences for supporting this work (http://www.meteothink.org/, last access: 7 April 2023).

*Competing interests.*

The contact author has declared that neither they nor their co-author has any competing interests.

*Financial support.*

This research has been supported by the Natural Science Foundation of Jiangsu Province (grant no. BK20211158), the National Nature Science Foundation of China (grant no. 42275102), and the Basic Special Business Fund for R&D for the Central Level Scientific Research Institutes of Nanjing Institute of Environmental Sciences  (grant no. GYZX210501)

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
