# Peer review of "$_{ m 1}$ Quantifying the seasonal variations and regional transport of PM2.5"

_EGUsphere, 2023_

## Author Response (AR1)

**A point-to-point response to the referee's comments**

Reply to anonymous referee 1

On behalf of my co-authors, we would like to express our great appreciation for your constructive comments and great efforts on our manuscript entitled "Quantifying the seasonal variations and regional transport of PM$_{2.5}$ in the Yangtze River Delta region, China: Characteristics, sources, and health risks" (EGUSPHERE-2023-489). We have studied the constructive comments carefully and have revised our manuscript. Replies to comments are in blue, and the line number in response refers to the unmarked manuscript, in which all revisions have been accepted.

General comments: Particle pollution is of great concern in many areas of East Asia. The Yangtze River Delta region in China is one of these typical highly polluted areas. This paper investigates the source apportionment of PM$_{2.5}$ by applying positive matrix factorization based on a regional background site in the Yangtze River Delta region and pays attention to the health burden of PM$_{2.5}$ pollution. The topic and some findings are fascinating. This manuscript not only provides some phenomena but also is a good job of explaining the mechanism. In all, the manuscript can be considered to be published after minor revision. The specific comments are listed below:

Response: Thanks for the careful evaluation of this manuscript. All your comments and suggestions are critical. They have direct significance to our paper writing and research work.

Specific comments:

(1) The English should be checked and polished.

Response: Thanks for your suggestion. According to this suggestion, we checked and polished the manuscript with the aid of experienced researchers.

(2) In Introduction

On lines 52-53, "Air exposure models have been widely used to further assess the non-carcinogenic and carcinogenic health risks of toxic elements in PM$_{2.5}$". The source apportionment and health risk of PM$_{2.5}$ pollution is the focus of this study, and the health risk assessment of this study should conduct a brief description.

Response: Thanks for your suggestion. We have added an introduction to air exposure models and

references to the recent national and international literature on health risk assessment. Please see lines 57-61 in the revised manuscript.

The added references include:

Thurston, G. D., Burnett, R. T., Turner, M. C., Shi, Y., Krewski, D., Lall, R., Ito, K., Jerrett, M., Gapstur, S. M., Diver, W. R., and Pope III, C. A.: Ischemic heart disease mortality and long-term exposure to source-related components of US fine particle air pollution. Environ. Health Pers., 124(6), 785-794, https://doi.org/10.1289/ehp.1509777, 2016.

Conibear, L., Butt, E. W., Knote, C., Arnold, S. R., and Spracklen, D. V.: Residential energy use emissions dominate health impacts from exposure to ambient particulate matter in India. Nature communications, 9(1), 617, https://doi.org/10.1038/s41467-018-02986-7, 2018.

In the second and third paragraphs of the introduction section, the authors introduce the chemical components of $PM_{2.5}$ and the source apportionment methods in a scientific way. The authors summarize and analyze the shortcomings and advantages of relevant literature researches worldwide. However, more papers about air pollution in China should be added to support the ideas in the introduction.

Response: Thanks for your suggestion. The chemical composition of $PM_{2.5}$ and source apportionment methods are the focus of this paper. We have added more references (listed below) about air pollution in China to support the ideas in the introduction. Please see lines 65-69 in the revised manuscript.

Also, some papers about air pollution in China are added to support the ideas in the introduction, including

Fan, H., Zhao, C. F., and Yang, Y. K.: A comprehensive analysis of the spatio-temporal variation of urban air pollution in China during 2014-2018, Atmos. Environ., 220, https://doi.org/10.1016/j.atmosenv.2019.117066, 2020.

Nie, D. Y., Chen, M. D., Wu, Y., Ge, X. L., Hu, J. L., Zhang, K., and Ge, P. X.: Characterization of Fine Particulate Matter and Associated Health Burden in Nanjing, Int. J. Env. Res. Public Health, 15, https://doi.org/10.3390/ijerph15040602, 2018.

Zeng, Y. Y., Cao, Y. F., Qiao, X., Seyler, B. C., and Tang, Y.: Air pollution reduction in China: Recent success but great challenge for the future, Sci. Total Environ., 663, 329-337,

https://doi.org/10.1016/j.scitotenv.2019.01.262, 2019.

(3) In Data and Methodology

In Section 2.1, the author introduces detailed information on the sampling and measuring instruments. Whether there are parallel and blank sampling in this study. How about the distribution of pollution sources around the sampling site?

Response: Thanks for your comment. In the study, $PM_{2.5}$ component data was hourly collected, and the study was based on high-time resolution data. We measured 10% of all samples as parallel sampling and the pass rate was over 95%. The QA/QC procedures have passed the artificial random inspection of extreme value and time consistency. Please see lines 108-114 in the revised manuscript.

In the study, the $PM_{2.5}$ composition sampling site is on the rooftop of the School of Atmospheric Sciences, Xianlin Campus, Nanjing University (32.12 °N, 118.96 °E). The gaseous pollutant monitoring station is Xianlin Station (32.10 °N, 118.93 °E). There are no heavy industrial plants around the two sampling sites. Coal combustion, vehicle emission, and industrial sources have less impact on sampling sites than secondary inorganic aerosol sources.

On lines 132-133, "Second, we calculated the uncertainty (Unc) for each species based on the concentration fraction and MDL." How to set the uncertainty of different data? Detailed formulas on the Unc and MDL are lacking.

Response: Thanks for your suggestion. According to the EPA PMF 5.0 User Guide (US EPA, 2014), we added the detailed formulas of the Unc and MDL on lines 144-148 in the revised manuscript.

The added information is also listed as below:

$$"Unc = \frac{5}{6} \times MDL$$

$$Unc = \sqrt{\left(Error\ Fraction \times concentration\right)^2 + \left(0.5 \times MDL\right)^2}$$

where Unc is the uncertainty. MDL is the method detection limit. If the concentration is less than or equal to the MDL provided, Unc is calculated using a fixed fraction of the MDL (Taylor et al., 2020). If the concentration is greater than the MDL, the calculation is based on the concentration fraction and

MDL.".

On lines 156-157, "The CWT method divided the research area into small equal grids, set a standard value for the research object, and defined the trajectory exceeding the standard value as the pollution trajectory." It is necessary to present the standard value of the research object in this study. Why use this standard value?

Response: Thanks for your suggestion. The daily average concentration of 75 μg·m$^{-3}$ is the Grade II criteria of the Chinese National Ambient Air Quality Standards (NAAQS). Therefore, we used the standard value of 75 μg·m$^{-3}$ in this study. Please see lines 175-176 in the revised manuscript.

(4) In Results and discussions

On lines 247-248, "meteorological conditions provided favorable conditions for the transformation of gaseous precursors." There are many factors affecting the oxidation rate of gaseous precursors of PM$_{2.5}$. In addition to relative humidity, meteorological factors such as temperature, radiation intensity, and boundary layer height all have a significant impact on oxidation rates. Please further explain the reason. For example, comparing the correlation coefficients of PM$_{2.5}$ with the main meteorological factors in different seasons.

Response: Thanks for your comment. According to your suggestion, we added the correlation coefficients of PM$_{2.5}$ with the main meteorological factors on lines 282-285 in the revised manuscript.

The added lines are also shown below:

"In summer, the correlation coefficients of PM$_{2.5}$ with RH, T, WS, and BLH were 0.42, -0.47, -0.15, and -0.23, respectively. In winter, the correlation coefficients of PM$_{2.5}$ concentration with RH, T, WS, and BLH were 0.74, -0.57, -0.31, and -0.32, respectively. High RH (79.6%), low temperature (4.9°C), low WS (2.1m·s$^{-1}$), and low BLH (419.7m) provided favorable conditions for the accumulation of PM$_{2.5}$.".

On Line 261, please add the meaning of "C", "MP", and "HP" in Fig. 2.

Response: Thanks for your suggestion. We have already made the corresponding modification as "Chemical compositions of PM$_{2.5}$ and mass ratio of OC/EC at different pollution levels of the total

samples per season. C, MP, and HP represent the clean day, moderately polluted day, and heavily polluted day, respectively. "%" represents the proportion of the filter sample quantity at each pollution level out of the total samples.". For details, please see lines 303-305 in the revised manuscript.

On Lines 338-339, "However, the industrial pollution derived from long-range transport was attenuated by meteorological conditions." Which meteorological factors had the impacts? I think this needs to be explained in the text.

Response: Thanks for your comment. $PM_{2.5}$ pollution was more concentrated around the YRD region according to the CWT analysis. Based on the backward trajectory calculation, most of the long-range transport of $PM_{2.5}$ passed through the Yellow Sea and the East Sea. The oceanic air masses were influenced by tropical cyclones with high temperatures and strong winds, which had a great effect on mitigating $PM_{2.5}$ pollution. We have already made the corresponding modification and added references. Please see lines 396-399 of the revised manuscript.

The added references include:

Chow, W. S., Huang, X. H. H., Leung, K. F., Huang, L., Wu, X. R., and Yu, J. Z.: Molecular and elemental marker-based source apportionment of fine particulate matter at six sites in Hong Kong, China, Sci. Total Environ., 813, https://doi.org/10.1016/j.scitotenv.2021.152652, 2022.

Li, M., Hu, M., Guo, Q., Tan, T., Du, B., Huang, X., He, L., Guo, S., Wang, W., Fan, Y. and Xu, D.: Seasonal Source Apportionment of $PM_{2.5}$ in Ningbo, a Coastal City in Southeast China, Aerosol Air Qual. Res, 18: 2741-2752. https://doi.org/10.4209/aaqr.2018.01.0011, 2018.

Xu, H. M., Cao, J. J., Chow, J. C., Huang, R. J., Shen, Z., Chen, L. A., Ho, K. F. and Watson, J. G.: Inter-annual variability of wintertime $PM_{2.5}$ chemical composition in Xi'an, China: evidences of changing source emissions, Sci. Total Environ., 545, 546-555, https://doi.org/10.1016/j.scitotenv.2015.12.070, 2016.

In Section 3.4, the author investigated the non-carcinogenic and the carcinogenic risks in $PM_{2.5}$ and their total health risk in each source in Nanjing. The conclusion in this section should be compared with similar studies in other regions. Please briefly describe or cite some papers.

Response: Thanks for your comment. According to your suggestion, we added references to $PM_{2.5}$

health risks in other regions. Previous studies showed that coal combustion sources in Beijing, Shanxi, and Jinan had higher respiratory exposure and health risks, while the fugitive dust source in Liaoning contained higher levels of Pb, As, and Co (Hu et al., 2012; Zeng et al., 2019). However, toxic elements in $PM_{2.5}$ were within the acceptable level for a typical city in the YRD region in this study. For details, please see lines 418-422 of the revised manuscript.

The added references include:

Hu, X., Zhang, Y., Ding, Z. H., Wang, T. J., Lian, H. Z., Sun, Y. Y., and Wu, J. C.: Bioaccessibility and health risk of arsenic and heavy metals (Cd, Co, Cr, Cu, Ni, Pb, Zn and Mn) in TSP and $PM_{2.5}$ in Nanjing, China, Atmos. Environ., 57, 146-152, https://doi.org/10.1016/j.atmosenv.2012.04.056, 2012.

Zeng, Y. Y., Cao, Y. F., Qiao, X., Seyler, B. C., and Tang, Y.: Air pollution reduction in China: Recent success but great challenge for the future, Sci. Total Environ., 663, 329-337, https://doi.org/10.1016/j.scitotenv.2019.01.262, 2019.

(5) In Conclusions

On lines 375-377, "Based on values of $NO_3^-/SO_4^{2-}$ and OC/EC ratio, dominant vehicle emission occurred during the heavily polluted period, and the contribution of coal combustion increased in winter." The specific rate data should be given.

Response: Thanks for your suggestion. From summer to winter, the $NO_3^-/SO_4^{2-}$ ratio increased from 1.2 to 1.59. OC/EC ratio decreased by 56.1% from clean days to heavy pollution days. The average OC/EC ratio on heavy pollution days was 1.3. Both the increase in motor vehicle emissions and the formation of meteorological conditions conducive to pollutant accumulation contribute to the decrease in the OC/EC ratio. We have already made the corresponding modification. Please see lines 437-440 of the revised manuscript.

**A point-to-point response to the referee's comments**

Reply to anonymous referee 2

On behalf of my co-authors, we would like to express our great appreciation for your constructive comments and great efforts on our manuscript entitled "Quantifying the seasonal variations and regional transport of $PM_{2.5}$ in the Yangtze River Delta region, China: Characteristics, sources, and health risks" (EGUSPHERE-2023-489). We have studied the constructive comments carefully and have revised our manuscript. Replies to comments are in blue, and the line number in response refers to the unmarked manuscript, in which all revisions have been accepted.

General comments: This paper presents the source contribution of $PM_{2.5}$ in a heavily polluted area in China based on measurement of concentrations of PM components at high time resolution. Main source areas of the PM Factors identified are estimated by Backward trajectory analyses. Based on the concentrations measured of potentially toxic elements, carcinogenic and non-carcinogenic risk of these elements is estimated, identifying the sources with a higher potential impact on health.

Results presented in the manuscript are of high interest. In general, the methodology and interpretation of results is sound. However, the major issue of this paper is related to the writing. It should be thoroughly reviewed. Some parts must be reordered. The description should follow a clear order that facilitates understanding. many of the statements are not correctly augmented. There are missing references. Below I have a list of minor corrections (although it is not intended to be an exhaustive list).

Response: Thanks for the careful evaluation of this manuscript. All your comments and suggestions are critical. They have direct significance to our paper writing and research work.

Specific comments:

Abstract

Lines 23-24. It is confusing; you are discussing about emissions showing values of ambient concentrations. Moreover, what is the limit between high and moderate?

Response: Thanks for your comment. In the previous manuscript, we used pollutant concentrations to refer to pollutant emissions in the CWT analysis, which was clearly not rigorous. According to the suggestion, we have revised "emission" to "concentration" and only discussed the distribution of

pollutant concentrations. The threshold between high and moderate was redefined as 10 $\mu g \cdot m^{-3}$. The moderate concentration was between 5 and 10 $\mu g \cdot m^{-3}$ and the high concentration was defined as 10 $\mu g \cdot m^{-3}$ and more. Please see lines 28-29 in the revised manuscript.

Lines 25-27. Please, check sentence.

Response: Thanks for your comment. We have removed the sentence "The $PM_{2.5}$ pollution from long-range transport is attenuated by meteorological conditions and ocean air masses." in the revised manuscript.

Line 27. SIA cannot be considered as a source of pollution; SIA accounts for secondary inorganic compounds that can be emitted by a variety of sources. PMF grouped these components given to their major secondary origin and similar formation processes by transformation of precursor gases, mainly emitted by combustion.

Response: Thanks for your comment. In the previous manuscript, we defined the source with the highest proportion of the six sources obtained through the PMF model as "SIA". This created an ambiguity with the secondary inorganic compounds in $PM_{2.5}$. According to the suggestion, we have redefined this source as the "SIS" (secondary inorganic aerosol source) on line 25 of the revised manuscript and modified similar issues throughout the revised manuscript.

Introduction

Line 38. Check this value (35 $\mu g \cdot m^{-3}$). WHO 2006 AQG for $PM_{2.5}$ are 10 $\mu g \cdot m^{-3}$ as annual average concentration and 25 $\mu g \cdot m^{-3}$ as 24h mean concentration. Add references.

Response: Thanks for your comment. Since the annual average concentration of $PM_{2.5}$ was discussed in the references, we modified "35 $\mu g \cdot m^{-3}$" to "10 $\mu g \cdot m^{-3}$" based on the WHO 2006 AQG. The references about air quality guidelines of the World Health Organization have been added on lines 42-43.

The added references include:

Cheng, J., Tong, D., Zhang, Q., Liu, Y., Lei, Y., Yan, G., Yan, L., Yu, S., Cui, Y. R., Clarke, L., Geng, G. G., Zheng, B., Zhang, X. T., Davis, S. J., He, K. B.: Pathways of China's $PM_{2.5}$ air quality 2015－2060 in the context of carbon neutrality. National Science Review, 8(12), nwab078,

https://doi.org/10.1093/nsr/nwab078, 2021.

Song, C. B., He, J. J., Wu, L., Jin, T. S., Chen, X., Li, R. P., Ren, P. P., Zhang, L., and Mao, H. J.: Health burden attributable to ambient $PM_{2.5}$ in China. Environ., Pollut., 223, 575-586, https://doi.org/10.1016/j.envpol.2017.01.060, 2017.

Zeng, Y. Y., Cao, Y. F., Qiao, X., Seyler, B. C., and Tang, Y.: Air pollution reduction in China: Recent success but great challenge for the future, Sci. Total Environ., 663, 329-337, https://doi.org/10.1016/j.scitotenv.2019.01.262, 2019.

Lines 44-47. Please, check sentence.

Response: Thanks for your comment. Organic carbon (OC) comprises thousands of organic compounds. Elemental carbon (EC) is stable and mainly derived from primary sources of combustion products. We have already made the corresponding modification. Please see lines 50-52 in the revised manuscript.

Line 58-59. This sentence can be deleted.

Response: Thanks for your suggestion. The sentence "Compared with other methods, the PMF model can provide both the source profiles and contributions of various sources as a model outcome (Li et al., 2020)." has been removed in the revised manuscript.

Lines 66-69. I suggest moving this sentence to Line 54.

Response: Thanks for your suggestion. The sentence "The YRD region is China's scientific research base and comprehensive transportation hub. The annual $PM_{2.5}$ concentration in the YRD has been reduced by 45.6% from 2016 to 2018. " has been moved to line 66 in the revised manuscript.

Line 69. Can you demonstrate it is high quality data? No QA/QC information is presented in section 2. In this paragraph I would indicate that the study is based in high time resolution data.

Response: Thanks for your comment. In the study, $PM_{2.5}$ component data was hourly collected and the study was based on high-time resolution data. $PM_{2.5}$ component data included the elemental carbon, organic carbon, 30 trace elements, and 8 soluble components in aerosols. The QA/QC procedures have passed the artificial random inspection of extreme value and time consistency. We added the QA/QC procedures on lines 108-114 of the revised manuscript.

Data and Methodology

2.1

Line 78. Metals are also inorganic; I would say: "…and 8 soluble components (…". Same for line 85.

Response: We totally agree with your suggestion. The sentence "…and the 8 inorganic components" has been modified to "and 8 soluble components" on line 89 of the revised manuscript. The sentence "The inorganic components sampling instrument" has been modified to "The soluble components sampling instrument" on line 96 of the revised manuscript.

Lines 76-89: IMPORTANT: please, provide some information about blank analysis and QA/QC.

Response: Thanks for your suggestion. We measured 10% of all samples as parallel sampling and the pass rate was over 95%. We defined the missing sampling of atmospheric pollutant data as -999 to facilitate PMF processing. The chemical mass reconstruction method was used to correct potential measurement errors, which was described in detail in Section 2.2. The QA/QC procedures have passed the artificial random inspection of extreme value and time consistency. For details, please see lines 108-114 in the revised manuscript.

Lines 90-92: were these pollutants measured at the same site where $PM_{2.5}$?

Response: Thanks for your comment. In this study, $PM_{2.5}$ components sampling data were collected on the rooftop of the School of Atmospheric Sciences, Xianlin Campus, Nanjing University (32.12 °N, 118.96 °E). Gaseous pollutant observation data, including $PM_{2.5}$, $PM_{10}$, $O_3$, $NO_2$, $SO_2$, and CO, were monitored by the National Environmental Monitoring Center (NEMC) of China. The monitoring station is Xianlin Station (32.10 °N, 118.93 °E). The latitude and longitude of the two sampling sites are close to the same.

2.2

Line 109: Do you discard the influence of sea spray? Sea spray contribution can be estimated from $Cl^-$ and $Na^+$, using ratios from literature (i.e Lide, D.R., 2005 (http://www.hbepnetbase.com). A major fraction of $Na^+$ and $Cl^-$, and a fraction of Ca, Mg, $SO_4^{2-}$ and K can be or marine origin.

Response: No, we did not discard the influence of sea spray. According to the PMF result, we defined

the source that accounted for 5.8% of the total $PM_{2.5}$ sources as fugitive dust (FD). FD contained a high proportion of $Na^+$, $Cl^-$, Ca, Mg, and K. Ti, Fe, and Mg are both common crustal elements that can represent the source of mineral dust. $Na^+$ and $Cl^-$ are closely related to marine sea spray. $K^+$ and $Ca^{2+}$ are considered to be significant tracers of biomass burning, which have obvious seasonal variations. (Tong et al., 2020; Silva et al., 2022). Previous studies showed that sea salt contributed 4.5±3.3% of all sources of $PM_{2.5}$ in Nanjing (Li et al., 2016; Yan et al., 2020). In this study, we illustrated that FD contained the influence of mineral dust, biomass burning, and sea spray, accounting for 5.8% of all sources of $PM_{2.5}$. Please see lines 334-338 in the revised manuscript.

The added references include:

Li, H. M., Wang, Q. G., Yang, M., Li, F. Y., Wang, J. H., Sun, Y. X., Wang, C., Wu H. F., and Qian, X.: Chemical characterization and source apportionment of $PM_{2.5}$ aerosols in a megacity of Southeast China, Atmos. Res., 288, https://doi.org/10.1016/j.atmosres.2016.07.005, 2016.

Silva, L. F., Schneider, I. L., Artaxo, P., Núñez-Blanco, Y., Pinto, D., Flores, É. M., Gómez-Plata, L., Ramírez, O., and Dotto, G. L.: Particulate matter geochemistry of a highly industrialized region in the Caribbean: Basis for future toxicological studies, Geos. Front, 13(1), 101-115, https://doi.org/10.1016/j.gsf.2020.11.012, 2022.

Tong, S. Y., Kong, L. D., Yang, K. J., Shen, J. D., Chen, L., Jin, S. Y., Wang, C., Sha, F., and Wang, L.: Characteristics of air pollution episodes influenced by biomass burning pollution in Shanghai, China, Atmos. Environ., 238, https://doi.org/10.1016/j.atmosenv.2020.117756, 2020.

Yan, Y., Zheng, Q., Yu, R. L., Hu, G. R., Huang, H. B., Lin, C. Q., Cui, J. Y., and Yan, Y.: Characteristics and provenance implications of rare earth elements and Sr-Nd isotopes in $PM_{2.5}$ aerosols and $PM_{2.5}$ fugitive dusts from an inland city of southeastern China, Atmos. Environ., 220, https://doi.org/10.1016/j.atmosenv.2019.117069, 2020.

2.3: Did you use EPA PMF v5.0 software?

Response: Yes, we used the EPA PMF v5.0 software. To clarify this, we added the software version number on lines 131-132 in the revised manuscript.

2.5: For the calculation of HQ and LCR, you need to know the average daily exposure concentration

(ECinh). For the calculation of ECinh you need to know GA, ET, EF, ED and AT. How do you estimate ET, EF, ED and AT? You refer to two papers but it should be clarified how these parameters were estimated in the text.

Response: Thanks for your suggestion. We added the detailed formulas of ET, EF, ED, and AT according to references. ET is the exposure time, 24 $h \cdot d^{-1}$; EF is the exposure frequency, 365 $d \cdot yr^{-1}$; ED is the exposure duration, 30 yr; and AT is the average exposure time, calculated by ED yr × 365 $d \cdot yr^{-1}$ × 24 $h \cdot d^{-1}$ for non-carcinogens and 70 yr × 365 $d \cdot yr^{-1}$ × 24 $h \cdot d^{-1}$ for carcinogens. Please see lines 199-201 in the revised manuscript.

Results

3.1

Line 191. What is the time resolution for the average $PM_{2.5}$ concentrations? Hourly? Daily? Please, indicate it.

Response: Thanks for your comment. The time resolution for $PM_{2.5}$ concentrations was the daily average in section 3.1. We added the indication on lines 211-212 in the revised manuscript.

Line 193. Seasonal variations are also related to change in emission rates; no only meteorological conditions.

Response: Thanks for your comment. We agree that seasonal variations are also related to changes in emission rates. Thus, we added some lines of emission influences and relevant references according to the suggestion.

The added texts are "Biomass emissions in most cities and industrial emissions in industrial cities contribute 7-27% to $PM_{2.5}$ mass in applicable cities (Tao et al., 2017). Coal consumption and population density have a significantly positive effect on $PM_{2.5}$ concentration (Zhou et al., 2018; Chow et al., 2022). The highest level of $PM_{2.5}$ in winter was due to coal consumption, lower temperatures (4.9°C), higher humidity (79.6%), and lower BLH (419.7m) than in summer.". Please see lines 218-222 of the revised manuscript.".

The added references include:

Tao, J., Zhang, L., Cao, J., and Zhang, R.: A review of current knowledge concerning $PM_{2.5}$ chemical composition, aerosol optical properties and their relationships across China, Atmos. Chem. Phys., 17, 9485–9518, https://doi.org/10.5194/acp-17-9485-2017, 2017.

Zhou, C. S., Chen, J., and Wang, S. J.: Examining the effects of socioeconomic development on fine particulate matter ($PM_{2.5}$) in China's cities using spatial regression and the geographical detector technique: Sci. Total Environ., 619, 436-445, https://doi.org/10.1016/j.scitotenv.2017.11.124, 2018.

Chow, W. S., Huang, X. H. H., Leung, K. F., Huang, L., Wu, X. R., and Yu, J. Z.: Molecular and elemental marker-based source apportionment of fine particulate matter at six sites in Hong Kong, China, Sci. Total Environ., 813, https://doi.org/10.1016/j.scitotenv.2021.152652, 2022.

Line 193. Please, delete "in spring" at the end of the line; it is duplicated.

Response: Thanks for your suggestion. The words "in spring" have been removed on line 215 in the revised manuscript.

Line 197. Height of the BLH also may influence on increasing PM concentrations in winter.

Response: Thanks for your comment. According to the suggestion, we added the value of BLH height and its correlation coefficient with the concentration of $PM_{2.5}$ in winter as follows: "Pearson correlation showed that $PM_{2.5}$ concentrations were significantly (p<0.01) correlated to WS (r=-0.36) in spring. In summer, high boundary layer height (BLH) (520.4m) significantly reduced $PM_{2.5}$ concentrations. In autumn and winter, $PM_{2.5}$ showed significant correlations between temperature (r=-0.53), relative humidity (r=0.62) and BLH (r=-0.43). High levels of $PM_{2.5}$ in winter were significantly affected by low temperature (4.9°C), high humidity (79.6%), and low BLH (419.7m).". For details, please see lines 215-218 in the revised manuscript

Line 202: replace "generally affected by vehicles" by "generally related to vehicle emissions"

Response: Thanks for your suggestion. The sentence "generally affected by vehicles" has been modified to be "generally related to vehicle emissions". Please see line 227 in the revised manuscript

Line 203: replace "normally affected by stationary sources" by "normally related to stationary sources"

Response: Thanks for your suggestion. The sentence "normally affected by stationary sources" has

been replaced as "normally related to stationary sources" on line 228 in the revised manuscript

Line 205: please, indicate which seasons correspond to the ratios

Response: Thanks for your suggestion. We have revised accordingly: "In this study, the average ratios of $NO_3^-/SO_4^{2-}$ were 1.81 in spring, 1.20 in summer, 2.34 in autumn, and 1.59 in winter, respectively.". Please see lines 230-232 in the revised manuscript.

Line 206: Ii is not possible to conclude that the contribution of mobile sources if greater than that of stationary sources based in the ratios $NO_3^-/SO_4^{2-}$. Actually, a significant fraction of NOx can also be emitted by stationary sources.

Response: Thanks for your comment. Since a significant fraction of NOx can also be emitted by stationary sources., it is not possible to conclude that the contribution of mobile sources is greater than that of stationary sources based in the ratios $NO_3^-/SO_4^{2-}$. Therefore, we further analyzed the oxidation rates of $SO_2$ and $NO_2$, and the OC/EC ratio in Section 3.2. The effects of motor vehicle, coal combustion, and biomass combustion sources on secondary aerosols are discussed separately from the perspective of inorganic and organic aerosols.

The modification of "The critical value of sulfur oxidation rate (SOR) and nitrogen oxidation rate (NOR) in the atmosphere are both 0.1 (Win et al., 2020). In this study, the order of the seasonal average NOR was winter (0.21) > spring (0.18) > autumn (0.17) > summer (0.15), while the order of the seasonal average SOR was winter (0.51) > spring (0.43) > autumn (0.42) > summer (0.36). $PM_{2.5}$ pollution in winter is associated with high RH and rapid production of particulate sulfate from the oxidation of $SO_2$ emitted by coal combustion (Wang et al., 2020). From summer to winter, the NOR and SOR values increased by 40.0% and 41.6%, respectively." is on lines 274-277 in the revised manuscript.

The modification of "The OC/EC mass ratio of motor vehicle emissions (1.1) is lower than that of coal combustion (2.7) and biomass burning (9.0) (Xu et al., 2021). In this study, the OC/EC ratios continuously decreased as air pollution got worse, and the values ranged from 6.1 (C), 4.1 (MP) to 3.9 (HP) in spring, from 6.2 (C) to 4.8 (MP) in autumn and from 4.3 (C), 2.7 (MP) to 1.3 (HP) in winter.

The annual average ratio of OC/EC decreased by 56.1% from clean days to heavy pollution days. If the OC/EC values were in the range of 2.5-5.0, vehicle exhaust emissions were considered the main source of OC and EC in aerosols, whereas if the OC/EC values were in the range of 5.0-10.5, coal combustion was considered the main source of OC and EC in aerosols (Gao et al., 2018; Liu et al., 2018). Distinct differences in the evolution of the OC/EC ratio on polluted days imply that mobile sources are likely more important. Both the increase in motor vehicle emissions and the formation of meteorological conditions conducive to pollutant accumulation contribute to the decrease in the OC/EC ratio." is on lines 292-301 of the revised manuscript.

Line 207: indicate that is % of PM

Response: Thanks for your suggestion. The sentence "...accounted for 12% and 14% in spring and winter" has been revised as "...accounted for 12% and 14% of $PM_{2.5}$ in spring and winter, respectively" on line 233 of the revised manuscript.

Lines 209-211: Please, add references.

Response: Thanks for your suggestion. We have added references of coal combustion and motor vehicle emission. Please see lines 236-237 in the revised manuscript.

The added references include:

Tao, J., Zhang, L., Cao, J., and Zhang, R.: A review of current knowledge concerning $PM_{2.5}$ chemical composition, aerosol optical properties and their relationships across China, Atmos. Chem. Phys., 17, 9485–9518, https://doi.org/10.5194/acp-17-9485-2017, 2017.

Jeong, C. H., Wang, J. M., Hilker, N., Debosz, J., Sofowote, U., Su, Y., Noble, M., Healy, R., Munoz, T., Celo, V., White, L., Audette, C., Herod, D., and Evans, G. J.: Temporal and spatial variability of traffic-related $PM_{2.5}$ sources: Comparison of exhaust and non-exhaust emissions, Atmos. Environ., 198, 55-69. https://doi.org/10.1016/j.atmosenv.2018.10.038, 2019

Caption and first row of Table 2. Delete "Average concentration $\mu g \cdot m^{-3}$ and component percentage %)" from Table and add this information in caption.

Improve caption: Seasonal average concentration of components of $PM_{2.5}$, in $\mu g \cdot m^{-3}$ and % in brackets,

and meteorological parameters.

Response: Thanks for your suggestion. "Average concentrations: µg·m$^{-3}$ (component percentages: %)" has been removed from Table 2, and this information is added in the caption. Following the suggestion, we have also improved the caption of Table 2, which is shown as follows.

**Table 2. Seasonal average concentration of components of PM$_{2.5}$, in µg·m$^{-3}$ and % in brackets, and meteorological parameters. T, RH, WS, and BLH represent air temperature, relative humidity, wind speed and boundary layer height, respectively.**

| Components and meteorological parameters | Spring | Summer | Autumn | Winter |
|---|---|---|---|---|
| PM$_{2.5}$ | 99.1 | 23.7 | 38.9 | 113.9 |
| SO$_4^{2-}$ | 20.5 (20.7) | 5.2 (21.9) | 7.3 (18.8) | 31.5 (27.7) |
| NO$_3^-$ | 16.9 (17.1) | 5.3 (22.4) | 9.8 (25.2) | 27.2 (23.9) |
| NH$_4^+$ | 15.1 (15.2) | 3.2 (13.5) | 7.1 (18.3) | 11.5 (10.1) |
| OM | 11.7 (11.8) | 1.6 (6.8) | 4.1 (10.5) | 11.0 (9.7) |
| EC | 2.3 (2.3) | 0.8 (3.4) | 1.6 (4.1) | 3.6 (3.2) |
| Mineral dust | 13.2 (13.3) | 2.3 (9.7) | 2.7 (6.9) | 8.7 (7.6) |
| Trace metals | 2.7 (2.7) | 0.5 (2.1) | 0.5 (1.3) | 1.6 (1.4) |
| Cl$^-$ | 2.7 (2.7) | 1.6 (6.8) | 0.8 (2.1) | 1.7 (1.5) |
| T (°C) | 18.8 | 27.6 | 19.4 | 4.9 |
| RH (%) | 86.5 | 58.2 | 73.1 | 79.6 |
| WS (m·s$^{-1}$) | 3.5 | 2.9 | 2.7 | 2.1 |
| BLH (m) | 469.7 | 520.4 | 443.6 | 419.7 |

For details, please see lines 239-240 in the revised manuscript.

Paragraph from Lines 214-230. This paragraph can be improved. Please, try to organize it following the same order for the description of seasons and components.

Response: Thanks for your suggestion. We reorganized the paragraph following the same order for the

description of seasons and components. Please see lines 242-258 of the revised manuscript.

Line 215: lowest PM$_{2.5}$ concentration in spring was recorded at 14:00h, not at 17:00h.

Response: Thanks for your comment. We have already made the corresponding modification as "In spring (Fig. 1a), the highest and lowest PM$_{2.5}$ concentrations were 143.6 μg·m$^{-3}$ at 7:00 and 94.8 μg·m$^{-3}$ at 14:00, respectively.". Please see lines 243-244 in the revised manuscript.

Figure 1 caption: Average diurnal variation of the concentrations of major chemical components of PM$_{2.5}$ per each season.

Response: Thanks for your comment. We have changed the caption of Figure 1 to be "Average diurnal variation of the concentrations of major chemical components of PM$_{2.5}$ per each season".

In addition, we have added the description of average diurnal variation of the concentrations of major chemical components of PM$_{2.5}$ per each season as follow:

"In spring, the concentration of SNA had obvious diurnal variations. From 6:00 to 18:00, the average concentration of NO$_3^-$ increased from 17.6 to 21.8 μg·m$^{-3}$, while the average concentration of SO$_4^{2-}$ decreased from 23.2 to 15.9 μg·m$^{-3}$. In summer, the maximum concentration difference of SNA between day and night was less than 10 μg·m$^{-3}$. In autumn, the concentration of SNA increased at night and decreased during the day. The maximum concentration difference was more than 20 μg·m$^{-3}$. In winter, from 18:00 to 23:00, the concentration of SNA increased from 74.5 μg·m$^{-3}$ to 108.7 μg·m$^{-3}$, with increasing rates of 8.5 μg·m$^{-3}$·h$^{-1}$.".

For details, please see lines 244-252 in the revised manuscript.

Lines 233-236: Why did you select these values? A concentration of 75 μg·m$^{-3}$ cannot be considered as clean.

Line 234:" …concentrations were <75…"

Response: Thanks for your comment. The daily average concentration of 75 μg·m$^{-3}$ is the Grade II criteria of the Chinese National Ambient Air Quality Standards (NAAQS). The daily average concentration of PM$_{2.5}$ below 35 μg·m$^{-3}$ is considered as clean. Therefore, we have modified the pollution levels as "In this study, it was defined as the clean day (C) when the daily average PM$_{2.5}$

concentrations were less than 35 μg·m$^{-3}$, the moderate pollution day (MP) when PM$_{2.5}$ concentrations were more than or equal to 35 μg·m$^{-3}$ and less than 150 μg·m$^{-3}$, and the heavy pollution day (HP) when PM$_{2.5}$ concentrations were more than or equal to 150 μg·m$^{-3}$.". For details, please see lines 265-267 in the revised manuscript.

Line 235: Figure 2a instead of Figure 3a.

Response: Thanks for your comment. "Figure 3a..." has been modified to "Figure 2a...". Please see line 267 of the revised manuscript.

Line 235: Mean of WSIIs? Water soluble inorganics…? Please, write.

Response: Thanks for your suggestion. We have added the words "The annual average concentration of the water-soluble inorganic ions (WSIIs) was 41.9 μg·m$^{-3}$, and accounted for 61.8% of PM$_{2.5}$" on lines 267-268 in the revised manuscript.

Line 236: do you mean ratios or concentrations?

Response: Thanks for your comment. The word "contributions" has been replaced by "ratios" on line 269 of the revised manuscript.

Lines 238-248. It is worthy to explain the variation of the ration NO$_3^-$/SO$_4^{2-}$ from spring to winter during the medium and high pollution episodes. IN spring and autumn, concentrations of nitrate are higher than concentration of sulphate during the high pollution episodes. However, in winter sulphate dominates with respect to nitrate. Probably this is related to the influence of coal combustion for heating. Importance of coal combustion in winter is stated in the conclusions but it should be more clearly detailed in the results section.

Response: Thanks for your suggestion. We couldn't agree more. In winter sulfate dominates with respect to nitrate. Sulfate concentrations are strongly influenced by coal combustion for heating. Therefore, we have cited more papers about the fast production of particulate sulfate from the oxidation of sulfur dioxide (SO$_2$) emitted by coal combustion, and added the words "The order of the seasonal average NOR was winter (0.21) > spring (0.18) > autumn (0.17) > summer (0.15), while the order of the seasonal average SOR was winter (0.51) > spring (0.43) > autumn (0.42) > summer (0.36). PM$_{2.5}$

pollution in winter is associated with high RH and rapid production of particulate sulfate from the oxidation of $SO_2$ emitted by coal combustion (Wang et al., 2020). From summer to winter, the NOR and SOR values increased by 40.0% and 41.6%, respectively. SOR and NOR showed a strong positive correlation with relative humidity, with a correlation coefficient of 0.53 and 0.61, respectively. The contribution of coal combustion varied between 30 and 57% of $PM_{2.5}$ in winter (Zhang et al., 2017). Under the conditions of high coal combustion emissions and high RH, the rapid oxidation of $SO_2$ occurred to produce sulfate.". For details, please see lines 274-280 in the revised manuscript.

The added references include:

Wang, J. F., Li, J. Y., Ye, J. H., et al.: Fast sulfate formation from oxidation of $SO_2$ by $NO_2$ and HONO observed in Beijing haze. Nat. Commun., 11(1), 2844, https://doi.org/10.1038/s41467-020-16683-x, 2020.

Zhang, Z. Z., Wang, W. X., Cheng, M. M., Liu, S. J., Xu, J., He, Y. J., and Meng, F.: The contribution of residential coal combustion to $PM_{2.5}$ pollution over China's Beijing-Tianjin-Hebei region in winter, Atmos. Environ., 159, 147-161, https://doi.org/10.1016/j.atmosenv.2017.03.054, 2017.

Lines 247-248: Height of boundary layer can also influence on the accumulation and pollutant favouring reactions.

Response: Thanks for your comment. The height of BLH also has a significant effect on the accumulation of $PM_{2.5}$ in winter. We have added the analysis for the effects of BLH as below.

"The sensitivity of $PM_{2.5}$ to surface temperature, wind speed, and boundary layer height is negative, while the sensitivity to relative humidity is positive. In winter, the correlation coefficients of $PM_{2.5}$ concentration with RH, T, WS, and BLH were 0.74, -0.28, -0.12, and -0.32, respectively. High RH (79.6%), low temperature (4.9°C), low wind speed (2.1m·s$^{-1}$), and low BLH(419.7m) provided favorable conditions for the accumulation of $PM_{2.5}$."
For details, please see lines 281-285 in the revised manuscript.

Line 251: "The differences in different seasons..." can be replaced by "The seasonal differences...".

Response: Thanks for your comment. According to the suggestion, "The differences in different

seasons..." has been replaced by "The seasonal differences..." on line 288 of the revised manuscript.

Line 254: OC is not formed by photochemical reactions. Some particulate organic compounds are formed by photochemical reactions or other transformation processes.

Response: Thanks for your comment. We have revised the introduction of OC and EC as "OC comprises thousands of organic compounds. EC is stable and mainly derived from primary sources of combustion products (Zhang et al., 2017; Wu et al., 2020).". Please see lines 291-292 in the revised manuscript.

The added references include:

Wu, X., Cao, F., Haque, M., Fan, M. Y., Zhang, S. C., and Zhang, Y. L.: Molecular composition and source apportionment of fine organic aerosols in Northeast China, Atmos. Environ., 239, https://doi.org/10.1016/j.atmosenv.2020.117722, 2020.

Zhang, Z. Z., Wang, W. X., Cheng, M. M., Liu, S. J., Xu, J., He, Y. J., and Meng, F.: The contribution of residential coal combustion to $PM_{2.5}$ pollution over China's Beijing-Tianjin-Hebei region in winter, Atmos. Environ., 159, 147-161, https://doi.org/10.1016/j.atmosenv.2017.03.054, 2017.

Lines 259-260: Please, explain which were the changes in emissions. This can be also related to the meteorological scenarios favouring accumulation of pollutants.

Response: Thanks for your comment. According to the suggestion, we have analyzed the variation in emissions based on the OC/EC ratio. The added texts are listed below:

"If the OC/EC values were in the range of 2.5-5.0, vehicle exhaust emissions were considered as the main source of OC and EC in aerosols, whereas if the OC/EC values were in the range of 5.0-10.5, coal combustion was considered the main source of OC and EC in aerosols (Gao et al., 2018; Liu et al., 2018). Distinct differences in the evolution of the OC/EC ratio on polluted days imply that mobile sources are likely more important. Both the increase in motor vehicle emissions and the formation of meteorological conditions conducive to pollutant accumulation contribute to the decrease in the OC/EC ratio.".For details, please see lines 296-301 in the revised manuscript.

The added references include:

Gao, J. J., Wang, K., Wang, Y., Liu, S. H., Zhu, C. Y., Hao, J. M., Liu, H. J., Hua, S. B., Tian, H. Z.: Temporal-spatial characteristics and source apportionment of PM$_{2.5}$ as well as its associated chemical species in the Beijing-Tianjin-Hebei region of China, Environ. Pollut., 233, 714-724, https://doi.org/10.1016/j.envpol.2017.10.123, 2018.

Liu, Z. R., Gao, W. K., Yu, Y. C., Hu, B., Xin, J. Y., Sun, Y., Wang, L. L., Wang, G. H., Bi, X. H., Zhang, G. H., Xu, H. H., Cong, Z. Y., He, J., Xu, J. S., and Wang, Y. S.: Characteristics of PM$_{2.5}$ mass concentrations and chemical species in urban and background areas of China: Emerging results from the CARE-China network. Atmos. Chem. Phys., 18(12), 8849-8871, https://doi.org/10.5194/acp-18-8849-2018, 2018.

Figure 2. Quality of Figure can be improved. Caption: please add at the end of caption: "…of the total samples per season"

Response: Thanks for your suggestion. As shown in the following figure, we have improved the quality of Figure 2, and added "…of the total samples per season" in the caption.

[Figure]

**Figure 2. Chemical compositions of PM$_{2.5}$ and mass ratio of OC/EC at different pollution levels of the total samples per season. C, MP, and HP represent clean day, moderately polluted day, and heavily polluted day, respectively. "%" represents the proportion of the filter sample quantity at each pollution level out of the total samples.**

For details, please see lines 302-305 in the revised manuscript.

Lines 265-266. Here you talk about sources and later about Factors. Please, link the number of factor and the sources.

Response: Thanks for your suggestion. In the revised manuscript, we have added an introduction of the relationship between factors and sources as "The number of factors in the PMF model corresponded to the number of sources of $PM_{2.5}$ in this study. When the number of factors was set to six, the fitting degree of the model calculation results was the highest.". Please see lines 310-311 of the revised manuscript.

Line 267: explain better the meaning of "other sources". I would say: Factor 6 was not cvleraly assigned to a source, a was attributed to the mis contribution of different sources; then, it was named as "other sources (OS)" (or urban mix UM).

Response: Thanks for your comment. Factor 6 was unidentified and could be affected by coal combustion, industrial processes, and biomass burning. In the absence of a clear designation of the source, Factor 6 was attributed to an erroneous contribution from a different source. According to the suggestion, we have revised the meaning of "other sources (OS)". Please see lines 338-340 in the revised manuscript.

Line 270: Please, explain meaning of %.

Response: Thanks for your comment. The meaning of % is the proportion of each chemical component in each source of $PM_{2.5}$. For example, the proportion of $NO_3^-$ of $PM_{2.5}$ in the secondary inorganic aerosol source (SIS), coal combustion (CC), industry source (IS), vehicle emission (VE), and fugitive dust was 61.9%, 16.9%, 4.0%, 6.3%, 10.9%, respectively. We have added the meaning of "%" on line 314 in the revised manuscript.

Line 274: I would replace "Factor 2 was relevant to CC" by "Given the source profile, Factor 2 was related to Coal combustion emissions".

Response: According to your suggestion, the sentence "Factor 2 was relevant to CC" has been modified to be "Given the source profile, Factor 2 was related to Coal combustion emissions". Please see lines 321-322 in the revised manuscript.

Line 275: Please, start the sentence indicating that you are now describing Factor 3. "Factor 3 (Figure 3c) was characterized by the association of…"

Response: Thanks for your suggestion. We have added the corresponding description of Factor 3 at the beginning of the sentence "Factor 3 (Figure 3c) was characterized by the association of heavy metal pollutants such as As (42.8%), Pb (33.8%), Cr (61.1%), Zn (58.9%), Cu (59.4%), Fe (38.3%), and Mn (40.1%).". Please see lines 322-324 in the revised manuscript.

Lines 278-279: This is not clear; please, rewrite it.

Response: Thanks for your comments. We have rewritten these sentences as "However, the percentage of OC was only 11.3%, while rates of Zn (58.9%) and Cu (59.4%) were higher in Factor 3 (Fig. 3c). Cu, Zn, and OC are used as tracers of a mixed source of traffic and industrial, and OC is the major pollutant in the vehicle exhaust (Wang et al., 2020). Compared to motor vehicle emissions, Factor 3 should be significantly influenced by industrial activities. Cu and Zn were mainly from industrial process sources.". Please see lines 324-328 of the revised manuscript.

Lines 280. Again; please, start naming the Factor you are talking about.

Response: Thanks for your suggestion. We have added the corresponding description of Factor 4 at the beginning of the sentence "Factor 4 (Figure 3d) was characterized by the association of vehicle emissions, with the high proportions of Ni (54.7%), V (80.5%), OC (55.4%), EC (79.8%), and $NO_3^-$ (20.3%).". Please see lines 328-329 in the revised manuscript.

Lines 280. V and Ni are usually tracers of heavy oil combustion (i.e. shipping emissions; see in 't Veld, et al., 2021. Science of the Total Environment, 795, art. no. 148728, DOI: 10.1016/j.scitotenv.2021.148728, and references therein). Please, add references for V and NI as tracers of vehicle emissions.

Response: Thanks for your suggestion. We have already made the modification of Factors and added the corresponding references. Please see lines 329-334 in the revised manuscript.

The added texts are listed as follow:

"VOCs and NOx released from vehicles were the precursors of the secondary organic compounds and nitrate in $PM_{2.5}$ and were important catalysts for increased atmospheric oxidation (Guevara et al., 2021). OC and EC are mainly from the vehicle exhaust, and Ni and V are usually tracers of heavy oil combustion (Wu et al., 2020; Veld., 2021). Factor 4 contained a high proportion of OC, EC, and $NO_3^-$, which could be considered as vehicle emission, while factor 4 contained Ni and V, which were also influenced by shipping emissions (Gao et al., 2018; Veld., 2021)."

The added references include:

Gao, J. J., Wang, K., Wang, Y., Liu, S. H., Zhu, C. Y., Hao, J. M., Liu, H. J., Hua, S. B., Tian, H. Z.: Temporal-spatial characteristics and source apportionment of $PM_{2.5}$ as well as its associated chemical species in the Beijing-Tianjin-Hebei region of China, Environ. Pollut., 233, 714-724, https://doi.org/10.1016/j.envpol.2017.10.123, 2018.

Guevara, M., Jorba, O., Soret, A., Petetin, H., Bowdalo, D., Serradell, K., Tena, C., van der Gon, H. D., Kuenen, J., Peuch, V. H., and Garcia-Pando, C. P.: Time-resolved emission reductions for atmospheric chemistry modelling in Europe during the COVID-19 lockdowns, Atmos. Chem. Phys., 21, 773-797, https://doi.org/10.5194/acp-21-773-2021, 2021.

Veld, M., Alastuey, A., Pandolfi, M., Amato, F., Perez, N., Reche, C., and Querol, X.: Compositional changes of $PM_{2.5}$ in NE Spain during 2009–2018: A trend analysis of the chemical composition and source apportionment, Sci. Total Environ., 795, 148728, https://doi.org/10.1016/j.scitotenv.2021.148728, 2021.

Wu, X., Cao, F., Haque, M., Fan, M. Y., Zhang, S. C., and Zhang, Y. L.: Molecular composition and source apportionment of fine organic aerosols in Northeast China, Atmos. Environ., 239, https://doi.org/10.1016/j.atmosenv.2020.117722, 2020.

Line 284: As shown in Figure 33, Factor 5…"

Response: Thanks for your suggestion. In the revised manuscript, "In Figure 3e, this factor had..." has been replaced by "As shown in Figure 3e, Factor 5 had..." on line 334.

Lines 285-287: Fe, Ti, Mn, K, and Ca (also Al, Si) are tracers of mineral dust (both natural or anthropogenic). K (mainly as $K^+$) is also a tracer of biomass burning.

Response: Thanks for your suggestion. According to the PMF model, Factor 5 had relatively high proportions of Fe (31.1%), Ti (78.2%), $K^+$ (55.8%), $Ca^{2+}$ (60.5%), and $Mg^{2+}$ (48.3%). Ti, Fe, and Mg are both common crustal elements that can represent the source of mineral dust (Yan et al., 2020). $K^+$ and $Ca^{2+}$ are considered to be significant tracers of biomass burning (Tong et al., 2020; Silva et al., 2022). Therefore, Factor 5 was identified as a mixed source of mineral dust and biomass burning in this study. We have already made the modification of Factor 5 and added relevant references. Please see lines 334-338 in the revised manuscript.

The added references include:

Tong, S. Y., Kong, L. D., Yang, K. J., Shen, J. D., Chen, L., Jin, S. Y., Wang, C., Sha, F., and Wang, L.: Characteristics of air pollution episodes influenced by biomass burning pollution in Shanghai, China, Atmos. Environ., 238, https://doi.org/10.1016/j.atmosenv.2020.117756, 2020.

Silva, L. F., Schneider, I. L., Artaxo, P., Núñez-Blanco, Y., Pinto, D., Flores, É. M., Gómez-Plata, L., Ramírez, O., and Dotto, G. L.: Particulate matter geochemistry of a highly industrialized region in the Caribbean: Basis for future toxicological studies, Geos. Front, 13(1), 101-115, https://doi.org/10.1016/j.gsf.2020.11.012, 2022.

Yan, Y., Zheng, Q., Yu, R. L., Hu, G. R., Huang, H. B., Lin, C. Q., Cui, J. Y., and Yan, Y.: Characteristics and provenance implications of rare earth elements and Sr-Nd isotopes in PM2.5 aerosols and PM2.5 fugitive dusts from an inland city of southeastern China, Atmos. Environ., 220, https://doi.org/10.1016/j.atmosenv.2019.117069, 2020.

Figure 3. Caption: please, check it

Response: Thanks for your comment. The caption of Figure 3 has been revised to "Figure 3. Factor profile in each source for $PM_{2.5}$ during the whole year. The histograms are the mass concentration of each species to every species ($\mu g \cdot m^{-3}$), and the red dots are the relative contributions of each source to every species (%)." Please see lines 342-343 in the revised manuscript.

Paragraph from Lines 292 to 305. Please, carefully revise this paragraph. Do you have any explanation about the biomass burning source identification? Biomass was identified by other works in close areas but it was not identified in this study. Why? Was it due to the species analysed? Is BB included in other

sources?

Response: Thanks for your comment. According to the PMF model, Factor 5 had relatively high proportions of Fe (31.1%), Ti (78.2%), $K^+$ (55.8%), $Ca^{2+}$ (60.5%), and $Mg^{2+}$ (48.3%). Ti, Fe, and Mg are both common crustal elements that can represent the source of mineral dust (Yan et al., 2020). $K^+$ and $Ca^{2+}$ are considered to be significant tracers of biomass burning (Tong et al., 2020; Silva et al., 2022). Therefore, Factor 5 was identified as a mixed source of mineral dust and biomass burning in this study. We have already made the modification of Factor 5 and added relevant references. Please see lines 334-338 in the revised manuscript.

The added references include:

Tong, S. Y., Kong, L. D., Yang, K. J., Shen, J. D., Chen, L., Jin, S. Y., Wang, C., Sha, F., and Wang, L.: Characteristics of air pollution episodes influenced by biomass burning pollution in Shanghai, China, Atmos. Environ., 238, https://doi.org/10.1016/j.atmosenv.2020.117756, 2020.

Silva, L. F., Schneider, I. L., Artaxo, P., Núñez-Blanco, Y., Pinto, D., Flores, É. M., Gómez-Plata, L., Ramírez, O., and Dotto, G. L.: Particulate matter geochemistry of a highly industrialized region in the Caribbean: Basis for future toxicological studies, Geos. Front, 13(1), 101-115, https://doi.org/10.1016/j.gsf.2020.11.012, 2022.

Yan, Y., Zheng, Q., Yu, R. L., Hu, G. R., Huang, H. B., Lin, C. Q., Cui, J. Y., and Yan, Y.: Characteristics and provenance implications of rare earth elements and Sr-Nd isotopes in PM2.5 aerosols and $PM_{2.5}$ fugitive dusts from an inland city of southeastern China, Atmos. Environ., 220, https://doi.org/10.1016/j.atmosenv.2019.117069, 2020.

Line 298-299: Which are the local emission characteristics?

Response: Thanks for your comment. We have added the local emission characteristics and cited some references as "In the PRD, vehicle emissions, secondary nitrate, coal burning, and industrial emissions showed obvious local emission characteristics. An extra 30% $PM_{2.5}$ concentration was tightly related to local emissions in the downtown and industrial areas (Huang et al., 2014; Li et al., 2020; Chow et al., 2022).". Please see lines 350-352 in the revised manuscript.

The cited references include:

Chow, W. S., Huang, X. H. H., Leung, K. F., Huang, L., Wu, X. R., and Yu, J. Z.: Molecular and elemental marker-based source apportionment of fine particulate matter at six sites in Hong Kong,

China, Sci. Total Environ., 813, https://doi.org/10.1016/j.scitotenv.2021.152652, 2022.

Huang, X. F., Yun, H., Gong, Z. H., Li, X., He, L. Y.,    Zhang, Y. H., and Hu, M.: Source apportionment and secondary organic aerosol estimation of $PM_{2.5}$ in an urban atmosphere in China. Sci. China Earth Sci. 57, 1352–1362 (2014). https://doi.org/10.1007/s11430-013-4686-2, 2014.

Li, S. W., Chang, M. H., Li, H. M., Cui, X. Y., and Ma, L. Q.: Chemical compositions and source apportionment of $PM_{2.5}$ during clear and hazy days: Seasonal changes and impacts of Youth Olympic Games. Chem., 256, 127163, https://doi.org/10.1016/j.chemosphere.2020.127163, 2020.

Line 299. Delete "CC contributed 22.4%".

Response: Thank for your suggestion. In the revised manuscript, the sentence "CC contributed 22.4%..." has been removed.

Line 303: "in winter" is duplicated; please, delete once.
Response: Thanks for your suggestion. We have already made the corresponding modification as "For example, emissions from coal combustion in winter increased the contribution of CC significantly.". Please see line 356 in the revised manuscript

Table 3. Please add columns for each source identified (you can use the acronyms in the header) and present the % for each one; you don't need to write the name of source every time. It would be helpful if you present contribution for all sources identified, not only for the 3 main sources,
Response: Thanks for your suggestion. In the revised manuscript, we have already made the corresponding modification and presented contributions for all sources identified. The new Table 3 is on lines 359, and also listed below:

**Table 3. Comparisons of source apportionment among different cities.**

| Location | | Time | Main pollution sources (proportion) | Investigator |
|---|---|---|---|---|
| YRD | Nanjing | 2018 | SIS (42.5%); CC (22.4%); IS (17.3%); VE (10.7%); FD (5.8%) | This study |
| | Nanjing | 2015 | SIS (31.5%); CC (27.3%); Road dust (26.5%); Oil combustion (8.5%); IS (5.1%) | Li et al. (2020) |
| | Ningbo | 2015 | SIS (39.2%); VE (21.4%); CC (12.4%); IS (9.5%); Ship emission (7.4%); BB (5.1); Aged sea salt (3.7%) | Li et al. (2018) |

| | | | | | |
|---|---|---|---|---|---|
| BTH | Beijing | 2017 | SIS (35.6%); CC (30.8%); BB (17.6%); VE (12.4%); IS (6.3%) | | Xu et al. (2021) |
| | Tianjin | 2014 | SIS (29.2%); IS (28.2%); CC (12.4%); VE (11.7%); Dust (11.7%); BB (5.3%) | | Huang et al. (2017) |
| | Shijiazhuang | 2014 | SIS (36.4%); IS (27.3%); CC (15.5%); VE (8.5%); Dust (7.0%); BB (2.8%) | | Huang et al. (2017) |
| PRD | Hong Kong | 2015 | SIS (44.9%); IS (13.5%); BB (10.8%);VE (8.6%); Oil combustion (5.3%); Aged sea salt (2.1%) | | Chow et al. (2022) |
| | Shenzhen | 2014 | SIS (39.3%); VE (26.9%); BB (9.8%); Aged sea salt (4.7%); Dust (3.5) | | Huang et al. (2014) |
| | Guangzhou | 2014 | SIS (34.6%); VE (28.6%); BB (23.1%); CC (17.7%); Ship emission (14.0%); IS (4.7%) | | Li et al. (2020) |

Figure 4. It can be improved. Enlarge legend. Improve caption: Average annual contribution of the sources identified for PM$_{2.5}$ in Nanjing in 2018.

Response: Thanks for your suggestion. According to the suggestion, we have improved the quality of Figure 4. The caption has been made the corresponding modification. Please see line 360 in the revised manuscript and the following new Figure 4.

[Figure]

**Figure 4. Average annual contribution of the sources identified for PM$_{2.5}$ in Nanjing in 2018.**

Section 3.3.2 and Figures 5 and 6. It is difficult to follow the description and to understand the Figures if you don't know the location of Provinces and cities mentioned. Can you show them in the maps?

Response: Thanks for your suggestion. According to the suggestion, we have marked the capital cities of the provinces mentioned in section 3.3.2. Please see lines 400-402 in the revised manuscript and the following new figures.

[Figure]

**Figure 5. Source contributions to PM₂.₅ grouped by air masses associated with different 48-h backward trajectory clusters. The pie charts show the average source contribution for corresponding clusters.**

Lines 310-315. There is a clear increase of fugitive dust source during cluster a2 in Spring; what is the origin? Can be this related to long range transport of dust? Please, comment in the text.

Response: Thanks for your comment. Previous studies have shown that dust originating from the Gobi Desert can affect the YRD during frequent dust storms (Sun et al., 2018), while the YRD region was not affected by strong dust storms in 2018 (Wang et al., 2020). Therefore, this study focuses on the effect of industrial and construction activities on the fugitive dust source. In spring, the fugitive dust source contributed relatively highly in clusters a2 and a3, with proportions of 18.2% and 10.3%, respectively. Increased contribution from fugitive dust was related to industrial and construction activities. Cluster a2 originated in Liaoning and Cluster a3 was from northern Jiangxi. There were many industrial cities located in Liaoning, and the largest coal-fired thermal power plant in Jiangxi was located in the northern city of Jiujiang (Wang et al., 2019). Long-range transport of dust from these

areas would have a high impact on the formation of severe particle pollution in the YRD.We have already made the corresponding modification on lines 368-373 in the revised manuscript.

The cited references include:

Sun, T. Z., Che, H. Z., Qi, B., Wang, Y. Q., Dong, Y. S., Xia, X. G., Wang, H., Gui, K., Zheng, Y., Zhao, H. J., Ma Q. L., Du. R. G., and Zhang, X. Y.: Aerosol optical characteristics and their vertical distributions under enhanced haze pollution events: effect of the regional transport of different aerosol types over eastern China. Atmos. Chem. Phys., 18, 2949 – 2971, https://doi.org/10.5194/acp-18-2949-2018, 2018.

Wang, S. S., Hu, G. R., Yan, Y., Wang, S., Yu, R. L., and Cui, J. Y.: Source apportionment of metal elements in $PM_{2.5}$ in a coastal city in Southeast China: Combined Pb-Sr-Nd isotopes with PMF method, Atmos. Environ., 198, 302-312, https://doi.org/10.1016/j.atmosenv.2018.10.056, 2019.

Wang, Z. L., Huang, X., Wang. N., Xu J. W., and Ding, A. J.: Aerosol-Radiation Interactions of Dust Storm Deteriorate Particle and Ozone Pollution in East China. JGR. Atmos., https://doi.org/10.1029/2020JD033601, 2020.

Figure 5.- it can be improved. Difficult to see the number in the pies. I suggest decreasing the number figure for %. Caption: "… show the average source contribution…"

Response: Thanks for your suggestion. According to the suggestion, we have reduced the number figure for %. Also, we have modified the caption as "Figure 5. Source contributions to $PM_{2.5}$ grouped by air masses associated with different 48-h backward trajectory clusters. The pie charts show the average source contribution for corresponding clusters". Please see lines 400-402 in the revised manuscript.

Lines 323-324: $NH_4^+$ is an important tracer of agricultural activities. Please delete: "As a tracer of the biomass burning source".

Response: Thanks for your comment. According to the suggestion, "As a tracer of the biomass burning source..." has been removed in the revised manuscript.

Line 324: replace "Fig 7d" by "Figure 5d".

Response: Thanks for your suggestion. We have already replaced "Fig 7d" by "Figure 5d". Please see line 382 in the revised manuscript.

Line 328: replace "descending" by "ascending" (or by "increasing").

Response: Thanks for your suggestion. We have already replaced "descending" by "increasing". Please see line 385 in the revised manuscript.

Lines 338-339. I don' t understand the sentence about attenuation of the impact of long-range transport of industrial pollution by meteorology. This is not shown in Table 2. Please, explain better or deleted it.

Response: Thanks for your comment. According to the suggestion, we have removed this sentence

from the revised manuscript.

Lines 339-342. These 2 sentences can be removed.

Response: Thanks for your comment. According to the suggestion, we have removed these 2 sentences from the revised manuscript.

3.4. As afore mentioned, how the exposure parameters were estimated should be better explained. Authors refer to other papers, but this should be explained here; were calculated for this or using default values?

Response: Thanks for your suggestion. The exposure parameters and acceptable levels of the carcinogenic risk used in this study were default values. On line 414 of the revised manuscript, we have added some explanation and the reference as "The carcinogenic risk level of Cr ($1.0\times10^{-7}$) and As ($1.8\times10^{-5}$) were within the tolerance or acceptable level ($1\times10^{-6}$-$1\times10^{-4}$) (Zheng et al., 2019).".

Additionally, the detailed formulas of ET, EF, ED, and AT have been modified in section 2.5 as "ET is the exposure time, 24 h·d$^{-1}$; EF is the exposure frequency, 365 d·yr$^{-1}$; ED is the exposure duration, 30 year; and AT is the average exposure time, calculated by ED year $\times$ 365 d·yr$^{-1}$ $\times$ 24 h·d$^{-1}$ for non-carcinogens and 70 years $\times$ 365 d·yr$^{-1}$ $\times$ 24 h·d$^{-1}$ for carcinogens (Khan et al., 2016; Jiang et al., 2018).". Please see lines 199-201 in the revised manuscript.

The added references include:

Jiang, N., Duan, S. G., Yu, X., Zhang, R. Q., and Wang, K.: Comparative major components and health risks of toxic elements and polycyclic aromatic hydrocarbons of PM$_{2.5}$ in winter and summer in Zhengzhou: Based on three-year data, Atmos. Res., 213, 173-184, https://doi.org/10.1016/j.atmosres.2018.06.008, 2018.

Khan, M. F., Latif, M. T., Saw, W. H., Amil, N., Nadzir, M. S. M., Sahani, M., Tahir, N. M., and Chung, J. X.: Fine particulate matter in the tropical environment: monsoonal effects, source apportionment, and health risk assessment, Atmos. Chem. Phys., 16, 597-617, https://doi.org/10.5194/acp-16-597-2016, 2016.

Zheng, H., Kong, S. F., Yan, Q., Wu, F. Q., Cheng, Y., Zheng, S. R., Wu, J., Yang, G. W., Zheng, M. M., Tang, L. L., Yin, Y., Chen, K., Zhao, T. L., Liu, D. T., Li, S. L., Qi, S. H., Zhao, D. L., Zhang, T., Ruan, J. J., and Huang, M. Z.: The impacts of pollution control measures on PM$_{2.5}$ reduction: Insights of chemical composition, source variation and health risk, Atmos. Environ., 197, 103-117, https://doi.org/10.1016/j.atmosenv.2018.10.023, 2019.

Figure 7. Improve caption

Response: Thanks for your suggestion. We have already made the corresponding modification as "Figure 7. Non-carcinogenic (a) and carcinogenic (b) risks of toxic elements. Non-carcinogenic (c) and carcinogenic (d) risk of the sources identified for $PM_{2.5}$". Please see lines 428-429 in the revised manuscript.

Conclusions

Lines 375-377. As aforementioned this was not correctly discussed in the results section.

Response: Thanks for your comment. We have modified the conclusion about the $NO_3^-/SO_4^{2-}$ and OC/EC ratio as "From summer to winter, the $NO_3^-/SO_4^{2-}$ ratio increased from 1.2 to 1.59. The increase of NOR and SOR indicated that the secondary transformation of gaseous pollutants was strongly positively correlated with RH. OC/EC ratio decreased by 56.1% from clean days to heavy pollution days. The average OC/EC ratio on heavy pollution days was 1.3. Both the increase in motor vehicle emissions and the formation of meteorological conditions conducive to pollutant accumulation contribute to the decrease in the OC/EC ratio.". Please see lines 437-440 in the revised manuscript.

Data availability: is the $PM_{2.5}$ composition data available?

Response: Thanks for your comment. According to the suggestion, we have added the data availability of the $PM_{2.5}$ composition data on lines 455-460 of the revised manuscript.

The added texts are list below:

"$PM_{2.5}$ composition data were collected by the atmospheric heavy metal Monitor and the In-situ Gas and Aerosol Compositions Monitor in the School of Atmospheric Sciences, Nanjing University. Air quality monitoring data were acquired from the official NEMC real-time publishing platform (https://air.cnemc.cn:18007/, last access: 7 April 2023). Meteorological data were obtained from the University of Wyoming website (http://weather.uwyo.edu/, last access: 7 April 2023). The NCEP FNL data were taken from the NCEP (https://rda.ucar.edu/datasets/, last access: 7 April 2023). These data can be downloaded for free as long as one agrees to the official instructions."

---

## Author Response (AR2)

**A point-by-point response to the review**

On behalf of my co-authors, we would like to express our great appreciation for your constructive comments and great efforts on our manuscript entitled "Quantifying the seasonal variations and regional transport of $PM_{2.5}$ in the Yangtze River Delta region, China: Characteristics, sources, and health risks" (EGUSPHERE-2023-489). We have studied the constructive comments carefully and have revised our manuscript. Replies to comments are in blue, and the line number in response refers to the unmarked manuscript, in which all revisions have been accepted.

Lines 87-101. Explain how the aerosol particles were sampled for the different analysis methods. Did you uses a $PM_{2.5}$ selective inlet? Which filters did you use?

Response: Thanks for your comment. We have used the $PM_{2.5}$ selective inlet and filters in this article. The information is shown below:

In the monitoring of the trace elements, we used a particle cutting head to collect particles with an aerodynamic equivalent diameter of less than 100/10/2.5 μm in the ambient air, used organic microporous filter membranes to enrich the collected particles, used the principle of β-ray absorption to detect the concentration of particles enriched on the filter membranes and used the principle of X-ray Fluorescence to detect the concentration of more than 30 types of trace elements in the particles.

In the monitoring of the soluble components, aerosols were analyzed through the wetted inner and outer tubes, and water-soluble ions were absorbed in the absorbing liquid on the inner and outer walls of the tubes due to the diffusion principle and were carried out by the flushing. Finally, the collected water samples were removed from the bubbles and filtered and then imported into the Ion Layer Analyzer to analyze the composition of the gases dissolved in water.

We reorganized the paragraph following the same order for describing $PM_{2.5}$ compositions (OC, EC, trace elements, and soluble components). Please see lines 87-106 in the revised manuscript.

Lines 102-104. Please add information on the sampling method and location relative to yours.

Response: Thanks for your suggestion. In this study, Air pollutants, including $PM_{2.5}$, $PM_{10}$, $O_3$, $NO_2$, $SO_2$, and CO, were monitored by the National Environmental Monitoring Center (NEMC) of China. The nationwide observation network began operating in 74 major cities in 2013, and it included 1597 nonrural sites covering 454 cities by 2017. The monitoring Xianlin Station (32.10 °N, 118.93 °E) collected air pollutant data and automatically measured hourly air pollutants. These data were issued hourly on the national urban air quality real-time publishing platform (https://air.cnemc.cn:18007/, last access: 7 April 2023). We have added information on the sampling method and location. Please see lines 107-110 in the revised manuscript.

Line 146. "limi".

Response: Thanks for your comment. The word "limi" has been revised as "limit" on line 153 of the revised manuscript.

Lines 218-219. What do you mean with biomass emissions?

Response: Thanks for your comment. The term "biomass emissions" used in this article actually refers to "biomass burning". Biomass burning, in the form of open vegetation fires and indoor biofuel use, is one of the largest sources of many trace gases and aerosols in the global atmosphere (Andreae, et al., 2019). Gaseous pollutants, and even more so the particulate matter from biomass burning, pose grave risks to human health (Tao et al., 2017).

In this study, we cited biomass burning and industrial emissions data from the references to analyze the different proportions of various potential sources of $PM_{2.5}$ in urban areas. Therefore, we have changed the word "biomass emissions" to "biomass burning" and added relevant explanations and references. Please see lines 226-230 in the revised manuscript.

The added references include:

Andreae, M. O.: Emission of trace gases and aerosols from biomass burning an updated assessment, Atmos. Chem. Phys., 19, 8523–8546, https://doi.org/10.5194/acp-19-8523-2019, 2019.

Tao, J., Zhang, L., Cao, J., and Zhang, R.: A review of current knowledge concerning $PM_{2.5}$ chemical composition, aerosol optical properties and their relationships across China, Atmos. Chem. Phys., 17, 9485–9518, https://doi.org/10.5194/acp-17-9485-2017, 2017.

Line 239. Table 2. Please add the variability ranges for the values listed.

Response: Thanks for your suggestion. We have added the variability ranges for the compositions and meteorological parameters data on lines 247-248 of the revised manuscript.

**Table 2. Seasonal average concentration of components of $PM_{2.5}$, in µg·m$^{-3}$ and % in brackets, and meteorological parameters. T, RH, WS, and BLH represent air temperature, relative humidity, wind speed and boundary layer height, respectively.**

| Components and meteorological parameters | Spring | Summer | Autumn | Winter |
| --- | --- | --- | --- | --- |

| | | | | |
|---|---|---|---|---|
| PM$_{2.5}$ | 99.1 ± 29.5 | 23.7 ± 12.2 | 38.9 ± 20.6 | 113.9 ± 43.6 |
| SO$_4^{2-}$ | 20.5 ± 5.9 (20.7) | 5.2 ± 2.1 (21.9) | 7.3 ± 4.8 (18.8) | 31.5 ± 8.7 (27.7) |
| NO$_3^-$ | 16.9 ± 11.4 (17.1) | 5.3 ± 1.2 (22.4) | 9.8 ± 3.3 (25.2) | 27.2 ± 17.5 (23.9) |
| NH$_4^+$ | 15.1 ± 6.1 (15.2) | 3.2 ± 1.7 (13.5) | 7.1 ± 2.1 (18.3) | 11.5 ± 4.6 (10.1) |
| OM | 11.7 ± 6.1 (11.8) | 1.6 ± 0.7 (6.8) | 4.1 ± 1.1 (10.5) | 11.0 ± 5.8 (9.7) |
| EC | 2.3 ± 0.8 (2.3) | 0.8 ± 0.3 (3.4) | 1.6 ± 1.2 (4.1) | 3.6 ± 1.5 (3.2) |
| Mineral dust | 13.2 ± 4.5 (13.3) | 2.3 ± 0.8 (9.7) | 2.7 ± 1.0 (6.9) | 8.7 ± 2.7 (7.6) |
| Trace metals | 2.7 ± 1.5 (2.7) | 0.5 ± 0.1 (2.1) | 0.5 ± 0.2 (1.3) | 1.6 ± 0.9 (1.4) |
| Cl$^-$ | 2.7 ± 0.9 (2.7) | 1.6 ± 0.6 (6.8) | 0.8 ± 0.2 (2.1) | 1.7 ± 0.4 (1.5) |
| T (°C) | 18.8 ± 4.3 | 27.6 ± 5.4 | 19.4 ± 4.9 | 4.9 ± 2.2 |
| RH (%) | 86.5 ± 12.9 | 58.2 ± 6.3 | 73.1 ± 8.5 | 79.6 ± 10.4 |
| WS (m·s$^{-1}$) | 3.5 ± 0.6 | 2.9 ± 0.5 | 2.7 ± 0.5 | 2.1 ± 0.3 |
| BLH (m) | 469.7 ± 40.9 | 520.4 ± 58.9 | 443.6 ± 32.4 | 419.7 ± 23.5 |

Line 359. Table 3. This table is hard to read. Please consider converting it to a plot e.g. with bars.

Response: Thanks for your suggestion. We have converted Table 3 to Figure 4 (bar chart). Please see line 370 in the revised manuscript.

[Figure]

**Figure 4. Comparisons of source apportionment for PM$_{2.5}$ among different cities.**

Line 428. Figure 7. Explain the abbreviations on the figure in the caption e.g. HQ, LCR, SIS, IS, VM, FD, OS.

Response: Thanks for your comment. We have added the explanation of abbreviations (HQ, LCR, SIS, CC, IS, VM, FD, and OS) on the figure in the caption. Please see lines 442-446 in the revised manuscript.

[Figure]

**Figure 7. Non-carcinogenic (a) and carcinogenic (b) risks of toxic elements. Non-carcinogenic (c) and carcinogenic (d) risk of the sources identified for PM$_{2.5}$ in Nanjing. HQ, LCR, SIS, CC, IS, VM, FD, and OS represent hazard quotient, lifetime carcinogenic risk, secondary inorganic aerosol source, coal combustion, industry source, vehicle emission, fugitive dust, and other sources, respectively.**

Lines 455-456. Explain or give a link to were you make the results of your measurements available for other scientists.

Response: Thanks for your suggestion. The PM$_{2.5}$ composition data used in this article is measured by our team. We have added mailbox information to provide data. Please see lines 473-479 in the revised manuscript.

PM$_{2.5}$ composition data were collected by the atmospheric heavy metal Monitor and the In-situ Gas and Aerosol Compositions Monitor in the School of Atmospheric Sciences, Nanjing University (The data presented in this article are available upon request from Yangzhihao Zhan (zyzh1049744276@gmail.com)). Air quality monitoring data were acquired from the official NEMC real-time publishing platform (https://air.cnemc.cn:18007/, last access: 7 April 2023). Meteorological data were obtained from the University of Wyoming website (http://weather.uwyo.edu/, last access: 7 April 2023). The NCEP FNL data were taken from the NCEP (https://rda.ucar.edu/datasets/, last access: 7 April 2023). These data can be downloaded for free as long as one agrees to the official instructions.